# Imbalanced Semi-supervised Learning with Bias Adaptive Classifier

**Renzhen Wang[1], Xixi Jia[2], Quanziang Wang[1], Yichen Wu[3], Deyu Meng[1,4,5*]**
[1]Xi'an Jiaotong University, [2]Xidian University, [3]City University of Hong Kong
[4]Macau University of Science and Technology, [5]Peng Cheng Laboratory
{rzwang,dymeng}@mail.xjtu.edu.cn

## Abstract

Pseudo-labeling has proven to be a promising semi-supervised learning (SSL) paradigm. Existing pseudo-labeling methods commonly assume that the class distributions of training data are balanced. However, such an assumption is far from realistic scenarios and thus severely limits the performance of current pseudo-labeling methods under the context of class-imbalance. To alleviate this problem, we design a bias adaptive classifier that targets the imbalanced SSL setups. The core idea is to automatically assimilate the training bias caused by class imbalance via the bias adaptive classifier, which is composed of a novel bias attractor and the original linear classifier. The bias attractor is designed as a light-weight residual network and optimized through a bi-level learning framework. Such a learning strategy enables the bias adaptive classifier to fit imbalanced training data, while the linear classifier can provide unbiased label prediction for each class. We conduct extensive experiments under various imbalanced semi-supervised setups, and the results demonstrate that our method can be applied to different pseudo-labeling models and is superior to current state-of-the-art methods.

## 1 Introduction

Semi-supervised learning (SSL) (Chapelle et al., 2009) has proven to be promising for exploiting unlabeled data to reduce the demand for labeled data. Among existing SSL methods, pseudo-labeling (Lee et al., 2013), using the model's class prediction as labels to train against, has attracted increasing attention in recent years. Despite the great success, pseudo-labeling methods are commonly based on a basic assumption that the distribution of labeled and/or unlabeled data are class-balanced. Such an assumption is too rigid to be satisfied for many practical applications, as realistic phenomena always follows skewed distributions. Recent works (Hyun et al., 2020; Kim et al., 2020a) have found that class-imbalance significantly degrades the performance of pseudo-labeling methods. The main reason is that pseudo-labeling usually involves pseudo-label prediction for unlabeled data, and an initial model trained on imbalanced data easily mislabels the minority class samples as the majority ones. This implies that the subsequent training with such biased pseudo-labels will aggravate the imbalance of training data and further bias the model training.

To address the aforementioned issues, recent literature attempts to introduce pseudo-label re-balancing strategies into existing pseudo-labeling methods. Such a re-balancing strategy requires the class distribution of unlabeled data as prior knowledge (Wei et al., 2021; Lee et al., 2021) or needs to estimate the class distribution of the unlabeled data during training (Kim et al., 2020a; Lai et al., 2022). However, most of the data in imbalanced SSL are unlabeled and the pseudo-labels estimated by SSL algorithms are unreliable, which makes these methods sub-optimal in practice, especially when there are great class distribution mismatch between labeled and unlabeled data.

In this paper, we investigate pseudo-labeling SSL methods in the context of class-imbalance, in which class distributions of labeled and unlabeled data may differ greatly. In such a general scenario, the current state-of-the-art FixMatch (Sohn et al., 2020) may suffer from performance degradation. To illustrate this, we design an experiment where the entire training data (labeled data + unlabeled

---

*Corresponding author.

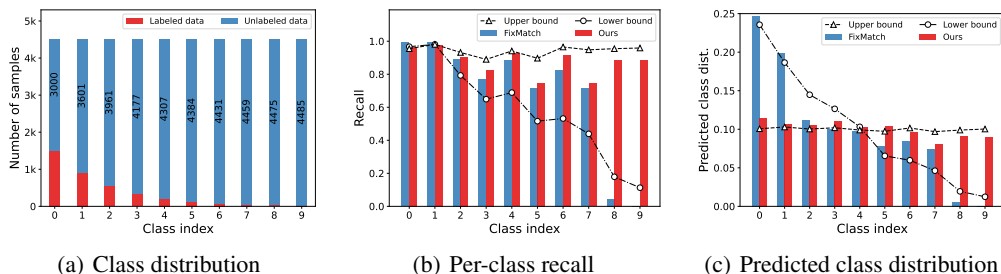

(a) Class distribution      (b) Per-class recall      (c) Predicted class distribution

Figure 1: Experiments on CIFAR-10-LT. (a) Labeled set is class-imbalanced with imbalance ratio $\gamma = 100$, while the whole training data remains balanced. Analysis on (b) per-class recall and (c) predicted class distribution for **Upper bound** (trained with the whole training data with ground truth labels), **Lower bound** (trained with the labeled data only), **FixMatch** and **Ours** on the balanced test set. Note that predicted class distribution is averaged by the predicted scores for all samples.

data) are balanced yet the labeled data are imbalanced, as shown in Fig. 1(a). Then, we can obtain an upper bound model by training the classification network on the whole training data (the underlying true labels of unlabeled data are given during training) and a lower bound model (trained with the labeled data only). As shown in Fig. 1(b)(c), the performance of FixMatch is much worse than the upper bound model, and degrades significantly from the majority classes to the tail classes , which instantiates the existence of imbalance bias. Meanwhile, for the last two tail classes, the performance of FixMatch is even worse than the lower bound model, which indicates the existence of pseudo-label bias brought by inaccurate pseudo-labels and it further deteriorates the model training.

To address this problem, we propose a *learning to adapt classifier* (L2AC) framework to protect the linear classifier of deep classification network from the training bias. Specifically, we propose a bias adaptive classifier which equips the linear classifier with a bias attractor (parameterized by a residual transformation). The linear classifier aims to provide an unbiased label prediction and the bias attractor attempts to assimilate the training bias arising from class imbalance. To this end, we learn the L2AC with a bi-level learning framework: the lower-level optimization problem updates the modified network with bias adaptive classifier over both labeled and unlabeled data for better representation learning; the upper-level problem tunes the bias attractor over an online class-balanced set (re-sampled from the labeled training data) for making the linear classifier predict unbiased labels. As a result, the bias adaptive classifier can not only fit the biased training data but also make the linear classifier generalize well towards each class (i.e., tend to equal preference to each class). In Fig. 1(c), we show that the linear classifier learned by our L2AC can well approximate the predicted class distribution of the upper bound model, indicating that L2AC obtains an unbiased classifier.

In summary, our contributions are mainly three-fold: (1) We propose to learn a bias adaptive classifier to assimilate online training bias arising from class imbalance and pseudo-labels. The proposed L2AC framework is model-agnostic and can be applied to various pseudo-labeling SSL methods; (2) We develop a bi-level learning paradigm to optimize the parameters involved in our method. This allows the online training bias to be decoupled from the linear classifier such that the resulting network can generalize well towards each class. (3) We conduct extensive experiments on various imbalanced SSL setups, and the results demonstrate the superiority of the proposed method. The source code is made publicly available at https://github.com/renzhenwang/bias-adaptive-classifier.

## 2 RELATED WORK

**Class-imbalanced learning** attempts to learn the models that generalize well to each classes from imbalanced data. Recent studies can be divided into three categories: Re-sampling (He & Garcia, 2009; Chawla et al., 2002; Buda et al., 2018; Byrd & Lipton, 2019) that samples the data to rearrange the class distribution of training data; Re-weighting (Khan et al., 2017; Cui et al., 2019; Cao et al., 2019; Lin et al., 2017; Ren et al., 2018; Shu et al., 2019; Tan et al., 2020; Jamal et al., 2020) that assigns weights for each class or even each sample to balance the training data; Transfer learning (Wang et al., 2017; Liu et al., 2019; Yin et al., 2019; Kim et al., 2020b; Chu et al., 2020; Liu et al., 2020; Wang et al., 2020) that transfers knowledge from head classes to tail classes. Besides, most recent works tend to decouple the learning of representation and classifier (Kang et al., 2020; Zhou et al., 2020; Tang et al., 2020; Zhang et al., 2021b). However, it is difficult to directly extend these

techniques to imbalanced SSL, as the distribution of unlabeled data is unknown and may be greatly different from that of labeled data.

**Semi-supervised learning** targets to learn from both labeled and unlabeled data, which includes two main lines of researches, namely pseudo-labeling and consistency regularization. Pseudo-labeling (Lee et al., 2013; Xie et al., 2020a;b; Sohn et al., 2020; Zhang et al., 2021a) is evolved from entropy minimization (Grandvalet & Bengio, 2004) and commonly trains the model using labeled data together with unlabeled data whose labels are generated by the model itself. Consistency regularization (Sajjadi et al., 2016; Tarvainen & Valpola, 2017; Berthelot et al., 2019b; Miyato et al., 2018; Berthelot et al., 2019a) aims to impose classification invariance loss on unlabeled data upon perturbations. Despite their success, most of these methods are based on the assumption that the labeled and unlabeled data follow uniform label distribution. When used for class-imbalance, these methods suffer from significant performance degradation due to the imbalance bias and pseudo-label bias.

**Imbalanced semi-supervised learning** has been drawing extensive attention recently. Yang & Xu (2020) pointed out that SSL can benefit class-imbalanced learning. Hyun et al. (2020) proposed a suppressed consistency loss to suppress the loss on minority classes. Kim et al. (2020a) introduced a convex optimization method to refine raw pseudo-labels. Similarly, Lai et al. (2022) estimated the mitigating vector to refine the pseudo-labels, and Oh et al. (2022) proposed to blend the pseudo-labels from the linear classifier with those from a similarity-based classifier. Guo & Li (2022) found a fixed threshold for pseudo-labeled sample selection biased towards head classes and in turn proposed to optimize an adaptive threshold for each class. Assumed that labeled and unlabeled data share the same distribution, Wei et al. (2021) proposed a re-sampling method to iteratively refine the model, and Lee et al. (2021) proposed an auxiliary classifier combined with re-sampling technique to mitigate class imbalance. Most recently, Wang et al. (2022) proposed to combine counterfactual reasoning and adaptive margins to remove the bias from the pseudo-labels. Different from these methods, this paper aims to learn an explicit bias attractor that could protect the linear classifier from the training bias and make it generalize well towards each class.

## 3 METHODOLOGY

### 3.1 PROBLEM SETUP AND BASELINES

Imbalanced SSL involves a labeled dataset $\mathcal{D}_l = \{(\mathbf{x}_n, \mathbf{y}_n)\}_{n=1}^N$ and an unlabeled dataset $\mathcal{D}_u = \{\mathbf{x}_m\}_{m=1}^M$, where $\mathbf{x}_n$ is a training example and $\mathbf{y}_n \in \{0, 1\}^K$ is its corresponding label. We denote the number of training examples of class $k$ within $\mathcal{D}_l$ and $\mathcal{D}_u$ as $N_k$ and $M_k$, respectively. In a class-imbalanced scenario, the class distribution of the training data is skewed, namely, the imbalance ratio $\gamma_l := \frac{\max_k N_k}{\min_k N_k} \gg 1$ or $\gamma_u := \frac{\max_k M_k}{\min_k M_k} \gg 1$ always holds. Note that the class distribution of $\mathcal{D}_u$, i.e., $\{M_k\}_{k=1}^K$ is usually unknown in practice. Given $\mathcal{D}_l$ and $\mathcal{D}_u$, our goal is to learn a classification model that is able to correctly predict the labels of test data. We denote a deep classification model $\Psi = f_\phi^{\mathrm{cls}} \circ f_\theta^{\mathrm{ext}}$ with the feature extractor $f_\theta^{\mathrm{ext}}$ and the linear classifier $f_\phi^{\mathrm{cls}}$, where $\theta$ and $\phi$ are the parameters of $f_\theta^{\mathrm{ext}}$ and $f_\phi^{\mathrm{cls}}$, respectively, and $\circ$ is function composition operator.

With pseudo-labeling techniques, current state-of-the-art SSL methods (Xie et al., 2020b; Sohn et al., 2020; Zhang et al., 2021a) generate pseudo-labels for unlabeled data to augment the training dataset. For unlabeled sample $\mathbf{x}_m$, its pseudo-label $\hat{\mathbf{y}}_m$ can be a 'hard' one-hot label (Lee et al., 2013; Sohn et al., 2020; Zhang et al., 2021a) or a sharpened 'soft' label (Xie et al., 2020a; Wang et al., 2021). The model is then trained on both labeled and pseudo-labeled samples. Such a learning scheme is typically formulated as an optimization problem with objective function $\mathcal{L} = \mathcal{L}_l + \lambda_u \mathcal{L}_u$, where $\lambda_u$ is a hyper-parameter for balancing labeled data loss $\mathcal{L}_l$ and pseudo-labeled data loss $\mathcal{L}_u$. To be more specific, $\mathcal{L}_l = \frac{1}{|\hat{\mathcal{D}}_l|} \sum_{\mathbf{x}_n \in \hat{\mathcal{D}}_l} \mathrm{H}\left(\Psi(\mathbf{x}_n), \mathbf{y}_n\right)$, where $\hat{\mathcal{D}}_l$ denotes a batch of labeled data sampled from $\mathcal{D}_l$, H is cross-entropy loss; $\mathcal{L}_u = \frac{1}{|\hat{\mathcal{D}}_u|} \sum_{\mathbf{x}_m \in \hat{\mathcal{D}}_u} \mathbb{1}(\max(p_m) \geq \tau) \mathrm{H}\left(\Psi(\mathbf{x}_m), \hat{\mathbf{y}}_m\right)$, where $p_m = \mathrm{softmax}(\Psi(\mathbf{x}_m))$ represents the output probability, and $\tau$ is a predefined threshold for masking out inaccurate pseudo-labeled data. For simplicity, we reformulate $\mathcal{L}$ as

$$\mathcal{L} = \frac{1}{|\hat{\mathcal{D}}_l|} \sum_{\mathbf{x}_i \in \hat{\mathcal{D}}_l} \mathrm{H}(\Psi(\mathbf{x}_i), \mathbf{y}_i) + \frac{1}{|\hat{\mathcal{D}}_u|} \sum_{\mathbf{x}_i \in \hat{\mathcal{D}}_u} \lambda_i \mathrm{H}(\Psi(\mathbf{x}_i), \hat{\mathbf{y}}_i), \tag{1}$$

where $\lambda_i = \lambda_u \mathbb{1}(\max(p_i) \geq \tau)$. The pseudo-labeling framework has achieved remarkable success in standard SSL scenarios. However, under class-imbalanced setting, the model is easily biased

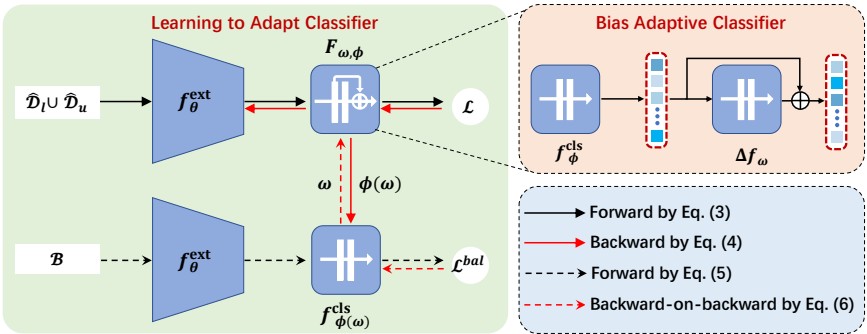

Figure 2: Learning to adapt classifier for imbalanced SSL. **Left:** The schematic illustration of the bi-level learning framework of our method, which includes four main steps: i) Forward process to compute the lower level loss in Eq. (3); ii) Backward process to update the classification network parameters $\phi(\omega)$, where $\omega$ are variables used for parameterizing $\phi$; (iii) Forward process to compute the upper-level loss in Eq. (5); iv) Backward-on-backward to update $\omega$. **Right:** The proposed bias adaptive classifier, which consists of the original linear classifier $f_\phi^{\mathrm{cls}}$ and a bias attractor $\Delta f_\omega$.

during training, such that the generated pseudo-labels can be even more biased and severely degrades the performance of minority classes. Moreover, due to the confirmation bias issue (Tarvainen & Valpola, 2017; Arazo et al., 2020), the model itself is hard to rectify such a training bias.

## 3.2 LEARNING TO ADAPT CLASSIFIER

Our goal is to enhance the existing pseudo-labeling SSL methods by making full use of both labeled and unlabeled data, while protecting the linear classifier from the training bias (imbalance bias and pseudo-label bias). To this end, we design to learn a bias adaptive classifier that equips the linear classifier with a bias attractor in order to assimilate complicated training bias.

**The proposed bias adaptive classifier:** As shown in Fig. 2. The bias adaptive classifier (denoted as $F$) consists of two modules: the linear classifier $f_\phi^{\mathrm{cls}}$ and a nonlinear network $\Delta f_w$ (dubbed bias attractor). The bias attractor is implemented by imposing a residual transformation on the output of the linear classifier, i.e., plugging $\Delta f_w$ after $f_\phi^{\mathrm{cls}}$ and then bridging their outputs with a shortcut connection. Mathematically, the bias adaptive classifier $F$ can be formulated as

$$F_{\omega,\phi}(\mathbf{z}) = (\mathbf{I} + \Delta f_\omega) \circ f_\phi^{\mathrm{cls}}(\mathbf{z}),  \tag{2}$$

where $\mathbf{z} = f_\theta^{\mathrm{ext}}(\mathbf{x}) \in \mathbb{R}^d$, $\mathbf{I}$ denotes identity mapping, and $\Delta f_w$ with parameters $\omega$ is a multi-layer perceptron (MLP) with one hidden layer in this paper. We design such a bias adaptive classifier for the following two considerations. On the one hand, the bias attractor $\Delta f_w$ adopts a nonlinear network which can assimilate complicated training bias in theory due to the universal approximation properties [1]. Since the whole bias adaptive classifier (i.e., classifier with the bias attractor) is required to fit the imbalanced training data, we hope the bias attractor could indeed help the linear classifier to learn the unbiased class conditional distribution (i.e., let the classifier less effected by the biases.). By contrast, in the original classification network, a single linear classifier is required to fit biased training data such that it is easily misled by class imbalance and biased pseudo-labels. On the other hand, the residual connection conveniently makes the bias attractor a plug-in module, i.e., assimilate the training bias during training and be removed in the test stage, and it also has been proven successful in easing the training of deep networks (He et al., 2016; Long et al., 2016).

**Learning bias adaptive classifier:** With the proposed bias adaptive classifier $F_{\omega,\phi}$, the modified classification network can be formulated as $\hat{\Psi} = F_{\omega,\phi} \circ f_\theta^{\mathrm{ext}}$. To make full use of the whole training data $(\mathcal{D}_l \cup \mathcal{D}_u)$ for better representation learning, we can minimize the following loss function:

$$\mathcal{L} = \frac{1}{|\hat{\mathcal{D}}_l|} \sum_{\mathbf{x}_i \in \hat{\mathcal{D}}_l} \mathrm{H}(\hat{\Psi}(\mathbf{x}_i), \mathbf{y}_i) + \frac{1}{|\hat{\mathcal{D}}_u|} \sum_{\mathbf{x}_i \in \hat{\mathcal{D}}_u} \lambda_i \mathrm{H}(\hat{\Psi}(\mathbf{x}_i), \hat{\mathbf{y}}_i).  \tag{3}$$

---

[1]In theory, an MLP can approximate almost any continuous function (Hornik et al., 1989).

This involves an optimization problem with respect to three parts of parameters $\{\theta, \phi, \omega\}$, which can be jointly optimized via the stochastic gradient decent (SGD) in an end-to-end manner. However, such a training strategy poses a critical challenge: there is no prior knowledge on $f_\phi^{\mathrm{cls}}$ to predict unbiased label prediction and on $\Delta f_w$ to assimilate the training bias. In other words, we cannot guarantee the training bias to be exactly decoupled from the linear classifier $f_\phi^{\mathbf{cls}}$. To address this problem, we take $\omega$ as hyper-parameters associated with $\phi$ and design a bi-level learning algorithm to jointly optimize the network parameters $\{\theta, \phi\}$ and hyper-parameters $\omega$.

We illustrate the process in Fig. 2 and Algorithm 1. In each training iteration $t$, we update the network parameters $\{\theta, \phi\}$ by gradient descent as

$$(\theta^{t+1}, \phi^{t+1}(\omega)) = (\theta^t, \phi^t) - \alpha \nabla_{\theta, \phi} \mathcal{L}, \tag{4}$$

where $\alpha$ is the learning rate. Note that we herein assume that $\omega$ is only directly related to the linear classifier $f_\phi^{\mathrm{cls}}$ via $\phi^{t+1}(\omega)$, which implies that the subsequent optimization of $\omega$ will not affect the feature extractor $f_\theta^{\mathrm{ext}}$. We then tune the hyper-parameters $\omega$ to make the linear classifier $\phi^{t+1}(\omega)$ generalize well towards each class, and we thus minimize the following loss function of the network $\Psi = f_{\phi^{t+1}(\omega)}^{\mathrm{cls}} \circ f_{\theta^{t+1}}^{\mathrm{ext}}$ over a class-balanced set (dynamically sampled from the labeled training set):

$$\mathcal{L}^{bal} = \frac{1}{|\mathcal{B}|} \sum_{\mathbf{x}_i \in \mathcal{B}} \mathrm{H}(f_{\phi^{t+1}(\omega)}^{\mathrm{cls}} \circ f_{\theta^{t+1}}^{\mathrm{ext}}(\mathbf{x}_i), \mathbf{y}_i), \tag{5}$$

where $\mathcal{B} \subset \mathcal{D}_l$ is a batch of class-balanced labeled samples, which can be implemented by class-aware sampling (Shen et al., 2016). This loss function reflects the effect of the hyper-parameter $\omega$ on making the linear classifier generalize well towards each class, we thus optimize $\omega$ by

$$\omega^{t+1} = \omega^t - \eta \nabla_\omega \mathcal{L}^{bal}, \tag{6}$$

where $\eta$ is the learning rate on $\omega$. Note that in Eq. (6) we need to compute a second-order gradient $\nabla_\omega \mathcal{L}^{bal}$ with respect to $\omega$, which can be easily implemented through popular deep learning frameworks such as Pytorch (Paszke et al., 2019) in practice.

In summary, the proposed bi-level learning framework ensures that 1) the linear classifier $f_\omega^{\mathrm{cls}}$ can fit unbiased class-conditional distribution by minimizing the empirical risk over balanced data via Eq. (5) and 2) the bias attractor $\Delta f_\omega$ can handle the implicit training bias by training the bias adaptive classifier $F_{\omega, \phi}$ over the imbalanced training data by Eq. (3). As such, our proposed $\Delta f_w$ can protect $f_w^{\mathrm{cls}}$ from the training bias. Additionally, our L2AC is more efficient than most existing bi-level learning methods (Finn et al., 2017; Ren et al., 2018; Shu et al., 2019). To illustrate this, we herein give a brief complexity analysis of our algorithm. Since our L2AC introduces a bi-level optimization problem, it requires one extra forward passes in Eq. (5) and backward pass in Eq. (6) compared to regular single-level optimization problem. However, in the backward pass, the second-order gradient of $\omega$ in Eq. (6) only requires to unroll the gradient graph of the linear classifier $f_\phi^{\mathrm{cls}}$. As a result, the backward-on-backward automatic differentiation in Eq. (6) demands a lightweight of overhead, i.e., approximately $\frac{\#\mathrm{Params}(f_\phi^{\mathrm{cls}})}{\#\mathrm{Params}(\Psi)} \times$ training time of one full backward pass.

### 3.3 THEORETICAL ANALYSIS

Note that the update of the parameter $\omega$ aims to minimize the problem Eq. (5), we herein give a brief debiasing analysis of our proposed L2AC by showing how the value of bias attractor $\Delta f(\cdot)$ change with the update of $\omega$. We use notation $\frac{\partial f}{\partial \omega}|_{\omega^t}$ to denote the gradient operation of $f$ at $\omega^t$ and superscript $T$ to denote the vector/matrix transpose. We have the following proposition.

**Proposition 3.1** *Let $\mathbf{p}_i$ denote the predicted probability of $\mathbf{x}_i$, then Eq. (6) can be rewritten as*

$$\omega^{t+1} = \omega^t + \eta \alpha \frac{1}{n} \sum_{i=1}^n G_i \frac{\partial \Delta f_i}{\partial \omega}|_{\omega^t}, \tag{7}$$

*where $G_i = \frac{\partial(\mathbf{p}_i - \mathbf{y}_i)}{\partial \phi}|_{\phi^t}^T \left( \frac{1}{m} \sum_{j=1}^m \frac{\partial \mathcal{L}_j^{\mathrm{bal}}(\theta, \phi)}{\partial \phi}|_{\phi^{t+1}} \right)$, which represents the similarity between the gradient of the sample $\mathbf{x}_i$ and the average gradient of the whole balanced set $\mathcal{B}$.*

This shows that the update of the parameter $\omega$ affects the value of the bias attractor, i.e, the value of $\Delta f_i$ is adjusted according to the interaction $G_i$ between $\mathbf{x}_i$ and $\mathcal{B}$. If $G_i > 0$ then $\Delta f_{i,k}$ is increased, otherwise $\Delta f_{i,k}$ is decreased, indicating L2AC adaptively assimilate the training bias.

Table 1: Comparison results on CIFAR-10 under two typical imbalanced SSL settings, i.e., $\gamma = \gamma_l = \gamma_u$ and $\gamma_l \neq \gamma_u$ ($\gamma_l = 100$). The performance (bACC / GM) is reported in the form of $mean_{\pm std}$ across three random runs.

| Methods | CIFAR-10 ($\gamma_l = \gamma_u$) | | CIFAR-10 ($\gamma_l \neq \gamma_u$) | |
| --- | --- | --- | --- | --- |
| | $\gamma = 100$ | $\gamma = 150$ | $\gamma_u = 1$ (uniform) | $\gamma_u = 100$ (reversed) |
| Vanilla | $58.8_{\pm 0.13}$ / $51.0_{\pm 0.11}$ | $55.6_{\pm 0.43}$ / $44.0_{\pm 0.98}$ | $58.8_{\pm 0.13}$ / $51.0_{\pm 0.11}$ | $58.8_{\pm 0.13}$ / $51.0_{\pm 0.11}$ |
| w/ Re-sampling | $55.8_{\pm 0.47}$ / $45.1_{\pm 0.30}$ | $52.2_{\pm 0.05}$ / $38.2_{\pm 1.49}$ | $55.8_{\pm 0.47}$ / $45.1_{\pm 0.30}$ | $55.8_{\pm 0.47}$ / $45.1_{\pm 0.30}$ |
| w/ LDAM-DRW | $62.8_{\pm 0.17}$ / $58.9_{\pm 0.60}$ | $57.9_{\pm 0.20}$ / $50.4_{\pm 0.30}$ | $62.8_{\pm 0.17}$ / $58.9_{\pm 0.60}$ | $62.8_{\pm 0.17}$ / $58.9_{\pm 0.60}$ |
| w/ cRT | $63.2_{\pm 0.45}$ / $59.9_{\pm 0.40}$ | $59.3_{\pm 0.10}$ / $54.6_{\pm 0.72}$ | $63.2_{\pm 0.45}$ / $59.9_{\pm 0.40}$ | $63.2_{\pm 0.45}$ / $59.9_{\pm 0.40}$ |
| MixMatch | $64.8_{\pm 0.28}$ / $49.0_{\pm 2.05}$ | $62.5_{\pm 0.31}$ / $42.5_{\pm 1.68}$ | $41.5_{\pm 0.76}$ / $12.0_{\pm 1.34}$ | $47.9_{\pm 0.09}$ / $20.5_{\pm 0.85}$ |
| w/ DARP | $67.9_{\pm 0.14}$ / $61.2_{\pm 0.15}$ | $65.8_{\pm 0.52}$ / $56.5_{\pm 2.08}$ | $86.7_{\pm 0.80}$ / $86.2_{\pm 0.82}$ | $72.9_{\pm 0.24}$ / $71.0_{\pm 0.32}$ |
| w/ SaR | $66.8_{\pm 0.92}$ / $59.9_{\pm 1.32}$ | $64.4_{\pm 2.21}$ / $57.3_{\pm 1.95}$ | $68.4_{\pm 3.20}$ / $62.0_{\pm 2.17}$ | $65.5_{\pm 1.01}$ / $64.2_{\pm 0.95}$ |
| w/ DASO | $69.8_{\pm 1.10}$ / $69.3_{\pm 1.07}$ | $66.5_{\pm 1.99}$ / $65.4_{\pm 2.25}$ | $75.5_{\pm 0.48}$ / $74.6_{\pm 0.67}$ | $65.7_{\pm 1.01}$ / $62.0_{\pm 1.23}$ |
| w/ ABC | $75.7_{\pm 0.76}$ / $74.7_{\pm 0.47}$ | $68.5_{\pm 0.40}$ / $56.4_{\pm 1.50}$ | $72.1_{\pm 0.53}$ / $41.2_{\pm 4.40}$ | $62.9_{\pm 0.36}$ / $59.9_{\pm 0.60}$ |
| w/ L2AC (ours) | $\mathbf{76.6}_{\pm 0.73}$ / $\mathbf{75.7}_{\pm 1.08}$ | $\mathbf{72.1}_{\pm 0.62}$ / $\mathbf{70.3}_{\pm 0.93}$ | $\mathbf{87.2}_{\pm 0.09}$ / $\mathbf{86.7}_{\pm 0.08}$ | $\mathbf{74.0}_{\pm 0.82}$ / $\mathbf{72.9}_{\pm 1.01}$ |
| FixMatch | $71.5_{\pm 0.72}$ / $66.8_{\pm 1.51}$ | $68.4_{\pm 0.15}$ / $59.9_{\pm 0.43}$ | $68.9_{\pm 1.95}$ / $42.8_{\pm 8.11}$ | $65.5_{\pm 0.05}$ / $26.0_{\pm 0.44}$ |
| w/ DARP | $75.5_{\pm 0.05}$ / $73.0_{\pm 0.09}$ | $70.4_{\pm 0.25}$ / $64.9_{\pm 0.17}$ | $85.4_{\pm 0.55}$ / $85.0_{\pm 0.65}$ | $74.9_{\pm 0.51}$ / $72.3_{\pm 1.13}$ |
| w/ CReST+ | $77.5_{\pm 0.15}$ / $76.1_{\pm 0.15}$ | $72.1_{\pm 0.74}$ / $68.9_{\pm 1.29}$ | N/A | N/A |
| w/ SaR | $77.6_{\pm 0.42}$ / $75.9_{\pm 0.76}$ | $71.5_{\pm 0.23}$ / $66.9_{\pm 0.25}$ | $85.9_{\pm 0.68}$ / $85.3_{\pm 0.53}$ | $78.3_{\pm 0.34}$ / $76.1_{\pm 0.21}$ |
| w/ DASO | $78.3_{\pm 0.55}$ / $76.5_{\pm 0.57}$ | $74.6_{\pm 0.74}$ / $71.7_{\pm 0.52}$ | $87.9_{\pm 0.41}$ / $87.7_{\pm 0.43}$ | $79.5_{\pm 0.91}$ / $78.9_{\pm 0.96}$ |
| w/ ABC | $80.2_{\pm 0.42}$ / $78.9_{\pm 1.29}$ | $74.7_{\pm 1.04}$ / $72.2_{\pm 1.45}$ | $81.3_{\pm 0.34}$ / $80.2_{\pm 0.36}$ | $70.3_{\pm 0.50}$ / $67.9_{\pm 0.70}$ |
| w/ L2AC (ours) | $\mathbf{82.1}_{\pm 0.57}$ / $\mathbf{81.5}_{\pm 0.64}$ | $\mathbf{77.6}_{\pm 0.53}$ / $\mathbf{75.8}_{\pm 0.71}$ | $\mathbf{89.5}_{\pm 0.18}$ / $\mathbf{89.2}_{\pm 0.19}$ | $\mathbf{82.2}_{\pm 1.23}$ / $\mathbf{81.7}_{\pm 1.36}$ |

## 4 EXPERIMENTS

We evaluate our approach on four benchmark datasets: CIFAR-10, CIFAR-100 (Krizhevsky et al., 2009), STL-10 (Coates et al., 2011) and SUN397 (Xiao et al., 2010), which are broadly used in imbalanced learning and SSL tasks. We adopt *balanced accuracy* (bACC) (Huang et al., 2016; Wang et al., 2017) and *geometric mean scores* (GM) (Kubat et al., 1997; Branco et al., 2016) [2] as the evaluation metrics. We evaluate our L2AC under two different settings: 1) both labeled and unlabeled data follow the same class distribution, i.e., $\gamma := \gamma_l = \gamma_u$; 2) labeled and unlabeled data have different class distributions, i.e., $\gamma_l \neq \gamma_u$, where $\gamma_u$ is commonly unknown.

### 4.1 RESULTS ON CIFAR-10

**Dataset.** We follow the same experiment protocols as Kim et al. (2020a). In detail, a labeled set and an unlabeled set are randomly sampled from the original training data, keeping the number of images for each class to be the same. Then both the two sets are tailored to be imbalanced by randomly discarding training images according to the predefined imbalance ratios $\gamma_l$ and $\gamma_u$. We denote the number of the most majority class within labeled and unlabeled data as $N_1$ and $M_1$, respectively, and we then have $N_k = N_1 \cdot \gamma_l^{\epsilon_k}$ and $M_k = M_1 \cdot \gamma_u^{\epsilon_k}$, where $\epsilon_k = \frac{k-1}{K-1}$. We initially set $N_1 = 1500$ and $M_1 = 3000$ following Kim et al. (2020a), and further ablate the proposed L2AC under various labeled ratios in Section 4.4. The test set remains unchanged and class-balanced.

**Setups.** The experimental setups are consistent with Kim et al. (2020a). Concretely, we employ Wide ResNet-28-2 (Oliver et al., 2018) as our backbone network and adopt Adam optimizer (Kingma & Ba, 2015) for 500 training epochs, each of which has 500 iterations. To evaluate the model, we use its exponential moving average (EMA) version, and report the average test accuracy of the last 20 epochs following Berthelot et al. (2019b). See Appendix D.1 for more details.

**Results under $\gamma_l = \gamma_u$.** We evaluate the proposed L2AC based on two widely-used SSL methods: MixMatch (Berthelot et al., 2019b) and FixMatch (Sohn et al., 2020), and compare it with the following methods: 1) The Vanilla model merely trained with labeled data; 2) Recent re-balancing methods that are trained with labeled data by considering class imbalance, including: Re-sampling (Japkowicz, 2000), LDAM-DRW (Cao et al., 2019) and cRT (Kang et al., 2020); 3) Recent imbalanced SSL methods, including: DARP (Kim et al., 2020a), CReST+ (Wei et al., 2021), ABC (Lee et al., 2021), SaR (Lai et al., 2022) and DASO (Oh et al., 2022). Please refer to Appendix C for more details about these methods. The main results are shown in Table 1. It can be observed that L2AC significantly improves MixMatch and FixMatch at least 9% absolute gain on bACC and at least 14%

---

[2] bACC and GM are defined as the arithmetic and geometric mean over class-wise sensitivity, respectively.

Table 2: Comparison results on CIFAR-100 and STL-10 under two different imbalance ratios. The performance (bACC / GM) is reported in the form of $mean_{\pm std}$ across three random runs.

| Methods | CIFAR-100 ($\gamma_l = \gamma_u$) | | STL-10 ($\gamma_u$ =N/A) | |
| --- | --- | --- | --- | --- |
| | $\gamma_l = 10$ | $\gamma_l = 20$ | $\gamma_l = 10$ | $\gamma_l = 20$ |
| FixMatch | $55.1_{\pm 0.09}$ / $46.7_{\pm 0.53}$ | $49.5_{\pm 0.38}$ / $34.2_{\pm 1.01}$ | $69.6_{\pm 0.60}$ / $62.6_{\pm 1.11}$ | $65.5_{\pm 0.05}$ / $26.0_{\pm 0.44}$ |
| w/ DARP | $56.3_{\pm 0.25}$ / $48.2_{\pm 0.73}$ | $50.2_{\pm 0.18}$ / $36.0_{\pm 0.60}$ | $72.9_{\pm 0.24}$ / $69.5_{\pm 0.18}$ | $74.9_{\pm 0.51}$ / $72.3_{\pm 1.13}$ |
| w/ ABC | $58.2_{\pm 0.08}$ / $51.8_{\pm 0.25}$ | $\mathbf{53.1}_{\pm 0.19}$ / $42.2_{\pm 0.82}$ | $78.2_{\pm 0.35}$ / $77.3_{\pm 0.30}$ | $72.7_{\pm 0.08}$ / $70.6_{\pm 0.22}$ |
| w/ DASO | $\mathbf{58.3}_{\pm 0.39}$ / $51.4_{\pm 0.80}$ | $53.0_{\pm 0.27}$ / $39.5_{\pm 1.45}$ | $78.2_{\pm 0.63}$ / $77.4_{\pm 0.53}$ | $75.4_{\pm 0.81}$ / $74.4_{\pm 1.00}$ |
| w/ L2AC (ours) | $57.8_{\pm 0.19}$ / $\mathbf{52.1}_{\pm 0.31}$ | $52.6_{\pm 0.13}$ / $\mathbf{43.0}_{\pm 0.45}$ | $\mathbf{79.9}_{\pm 0.52}$ / $\mathbf{79.1}_{\pm 0.49}$ | $\mathbf{77.0}_{\pm 0.65}$ / $\mathbf{75.8}_{\pm 0.68}$ |

Table 3: Comparison results on large-scale SUN397. The performance (bACC / GM) is reported in the form of mean±std across three random runs.

| Methods | bACC/GM | Methods | bACC/GM |
| --- | --- | --- | --- |
| Vanilla | $38.3_{\pm 0.05}$ / $29.9_{\pm 0.08}$ | DARP (Kim et al., 2020a) | $45.5_{\pm 0.32}$ / $37.5_{\pm 0.04}$ |
| cRT (Kang et al., 2020)] | $39.3_{\pm 0.21}$ / $33.7_{\pm 0.37}$ | ABC (Lee et al., 2021) | $47.0_{\pm 0.26}$ / $39.2_{\pm 0.34}$ |
| FixMatch Sohn et al. (2020) | $44.9_{\pm 0.11}$ / $35.7_{\pm 0.66}$ | L2AC (ours) | $\mathbf{48.8}_{\pm 0.19}$ / $\mathbf{40.6}_{\pm 0.17}$ |

on GM for all settings. This implies that our L2AC benefits the two baselines by learning an unbiased linear classifier. Moreover, our L2AC consistently surpasses all the comparison methods over both evaluation metrics. Take the extremely imbalanced case of $\gamma = 150$ for example, compared with the second best comparison method, our L2AC achieves up to 3.6% bACC gain and 4.9% GM gain upon MixMatch, and around 3.0% bACC gain and 3.6% GM gain upon FixMatch.

**Results under $\gamma_l \neq \gamma_u$.** The class distribution of labeled and unlabeled data can be arguably different in practice. We herein simulate two typical scenarios following Oh et al. (2022), i.e., the unlabeled set follows an uniform class distribution ($\gamma_u = 1$) and a reversed long-tailed class distribution against the labeled data ($\gamma_u = 100$ (reversed)). Note that CReST+ (Wei et al., 2021) fails in this case as they require the class distribution of unlabeled data as prior knowledge for training. As shown in Table 1, L2AC can consistently improve both MixMatch and FixMatch by a large margin. An interesting observation is that the baselines MixMach and FixMatch under $\gamma_u = 1$ perform much worse than that under $\gamma_u = 100$, even if more unlabeled data are added for $\gamma_u = 1$. This is mainly because the models under imbalanced SSL setting have a strong bias to generate incorrect labels for the tail classes, which will impair the entire learning process. As a result, more unlabeled tail class samples under $\gamma_u = 1$ lead to more severe performance degradation. On the contrary, our L2AC can eliminate the influence of the training bias and predict high-quality pseudo-labels for unlabeled data, such that it achieves significant performance gain in both settings.

## 4.2 Results on CIFAR-100 and STL-10

**Dataset:** To make a more comprehensive comparison, we further evaluate L2AC on CIFAR-100 (Krizhevsky et al., 2009) and STL-10 (Coates et al., 2011). For CIFAR-100, we create labeled and unlabeled sets in the same way as described in Sec. 4.1 and set $N_1 = 150$ and $M_1 = 300$. STL-10 is a more realistic SSL task that has no distribution information for unlabeled data. In our experiments, we set $N_1 = 450$ to construct the imbalanced labeled set and adopt the whole unknown unlabeled set (i.e., $M = 100k$). It is worth noting that the unlabeled set of STL-10 is noisy as it contains samples that do not belong to any of the classes in the labeled set. **Results:** As shown in Table 2, L2AC achieves the best performance over GM and competitive performance over bACC compared to current state-of-the-art ABC (Lee et al., 2021) and DASO (Oh et al., 2022). This implies that our approach obtains a relatively balanced classification performance towards all the classes. While for SLT-10, a more realistic noisy dataset without distribution information for unlabeled data, L2AC significantly outperforms ABC and DASO on both bACC and GM, which demonstrates that it has greater potential to be applied in the practical SSL scenarios.

## 4.3 Results on Large-Scale SUN-397

**Dataset:** SUN397 (Xiao et al., 2010) is an imbalanced real-world scene classification dataset, which originally consists of 108,754 RGB images with 397 classes. Following the experimental setups in

Table 4: Performance (bACC / GM) on CIFAR10-LT and STL-10 under various label ratio $\beta$.

| CIFAR-10 ($\gamma_l = \gamma_u = 100$) | $\beta = 1$ | $\beta = 5$ | $\beta = 10$ | $\beta$=20 | $\beta = 30$ |
|---|---|---|---|---|---|
| FixMatch | 54.9 / 16.5 | 65.1 / 35.5 | 69.0 / 53.9 | 72.0 / 62.2 | 76.5 / 74.3 |
| w/ L2AC (ours) | 62.8 / 55.8 | 75.9 / 74.1 | 79.3 / 78.4 | 80.8 / 79.9 | 83.6 / 83.2 |
| STL-10 ($\gamma_l = 10, \gamma_u = $ N/A) | $\beta = 5$ | $\beta = 10$ | $\beta = 20$ | $\beta$=40 | $\beta = 60$ |
| FixMatch | 46.5 / 19.9 | 48.8 / 27.0 | 58.2 / 39.8 | 67.2 / 60.7 | 69.2 / 67.6 |
| w/ L2AC (ours) | 62.8 / 57.1 | 66.5 / 62.9 | 72.6 / 70.6 | 77.0 / 75.7 | 78.8 / 77.9 |

Kim et al. (2020a), we hold-out 50 samples per each class for testing because no official data split is provided. We then construct the labeled and unlabeled dataset according to $M_k/N_k = 2$. The comparison methods includes: Vanilla, cRT (Kang et al., 2020), FixMatch (Sohn et al., 2020), DARP (Kim et al., 2020a) and ABC (Lee et al., 2021). More training details are presented in Appendix D.2. **Results:** The experimental results are summarized in Table 3. Compared to the baseline FixMatch (Sohn et al., 2020), our proposed L2AC results in about 4% performance gain over all evaluation metrics, and outperforms all the SOTA methods. This further verifies the efficacy of our proposed method toward the real-world imbalanced SSL applications.

## 4.4 DISCUSSION

**What about the performance under various label ratios?** To answer this question, we vary the ratios of labeled data (denoted as $\beta$) on CIFAR-10 and STL-10 to evaluate the proposed method. For CIFAR-10, we define $\beta = N_1/(N_1 + M_1)$ and set the imbalance ratio $\gamma = 100$. For STL-10, since it does not provide annotations for unlabeled data, we re-sample the labeled set from labeled data by $\beta = N_1/500$ and set the imbalance ratio $\gamma_l = 10$. As shown in Table 4, our L2AC consistently improves the baseline across different amounts of labeled data on both CIFAR-10 and STL-10. For example, STL-10 with $\beta = 5\%$ contains very scarce labeled data, where only 25 and 3 labeled samples belong to the most majority and minority classes, respectively. In such an extremely biased scenario, our L2AC significantly improves FixMatch by around 16% over bACC and 37% over GM.

**How does L2AC perform on the majority/minority classes?** To explain the source of performance improvements, we further visualize the confusion matrices on the test set of CIFAR-10 with $\gamma = 100$. Noting that the diagonal vector of a confusion matrix represents per-class recall. As shown in Fig. 3, our L2AC provides a relatively balanced per-class recall compared with the baseline FixMatch (Sohn et al., 2020) and other imbalanced SSL methods, e.g., DARP (Kim et al., 2020a) and ABC (Lee et al., 2021). It can also be observed that FixMatch easily tends to misclassify the samples of the minority classes into the majority classes, while our L2AC largely alleviates this bias. These results reveal that our L2AC provides an unbiased linear classifier for the test stage.

**Could L2AC improve the quality of pseudo-labels?** Qualitative and quantitative experiment results have shown that the proposed L2AC can improve the performance of pseudo-labeling SSL methods under different settings. We attribute this to the fact that L2AC can generate unbiased pseudo-labels during training. To validate this, we show per-class recall of pseudo-labels for CIFAR-10 with $\gamma_l = \gamma_u = 100$ and $\gamma_l = 100, \gamma_u = 100$ (reversed) in Fig. 4. It is clear that L2AC significantly raises the final recall of the minority classes. Especially for the situation where the distribution of labeled and unlabeled data are severely mismatched, as shown in Fig. 4(b), our L2AC considerably improves the recall of the most minority class by around 60% upon FixMatch. Such a high-quality pseudo-label estimation probably benefits from a more unbiased classifier with a basically equal preference for each class.

**How about the learnt linear classifier and feature extractor?** We revisit the toy experiment in Section 1 where the whole class-balanced training set is used to train an unbiased upper bound model. Instead, the baseline FixMatch and our L2AC are trained under the standard imbalanced SSL setups, and we compare the predicted class distributions on the test set with that of the upper bound model. As shown in Fig. 1(c), the classifier learned by L2AC approximates the predicted class distribution of the upper bound model and much better than FixMatch, indicating that L2AC results in a relatively unbiased classifier. **For the feature extractor**, we further visualize the representations of training data through t-SNE (Van der Maaten & Hinton, 2008) on CIFAR-10 with $\gamma_l = 100, \gamma_u = $

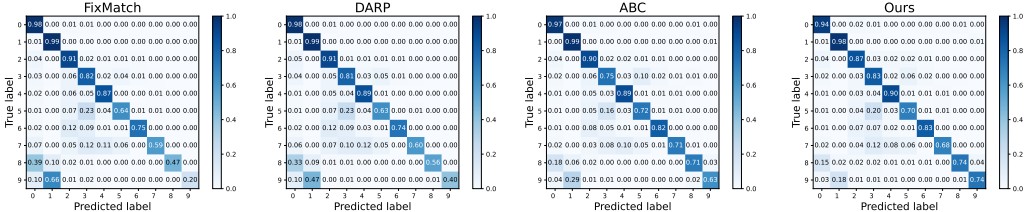

Figure 3: Confusion matrices of FixMatch (Sohn et al., 2020), DARP (Kim et al., 2020a), ABC (Lee et al., 2021), and ours on CIFAR-10 under the imbalance ratio $\gamma = 100$.

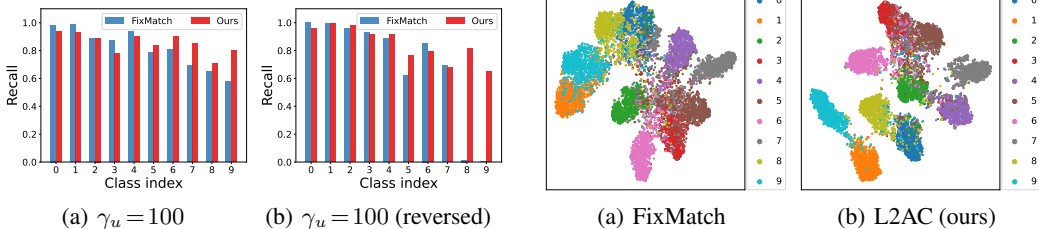

(a) $\gamma_u = 100$    (b) $\gamma_u = 100$ (reversed)    (a) FixMatch    (b) L2AC (ours)

Figure 4: Pseudo-label per-class recall of Fix-Match and ours on CIFAR-10 with $\gamma_l = 100$ and (a) $\gamma_u = 100$; (b) $\gamma_u = 100$ (reversed).

Figure 5: t-SNE visualization of training data for (a) FixMatch and (b) L2AC. L2AC helps to discriminate tail classes from majority ones.

1. As shown in Fig. 5, the features of the tail classes from FixMatch are scattered to the majority classes. However, L2AC can help the model to effectively discriminate the tail classes (e.g., Class 8, 9) from majority classes (e.g., Class 0, 1). Such a high-quality representation learning also benefits from an unbiased classifier during training. In Appendix E.1, we further present the unbiasedness of our L2AC by evaluating it on various imbalanced test sets.

**Ablation analysis.** We conduct an ablation study to explore the contribution of each critical component in L2AC. We experiment with FixMatch on CIFAR-10 under $\gamma_l = \gamma_u = 100$ and $\gamma_l = 100, \gamma_u = 100$ (reversed). **1)** We first verify whether the bias attractor is helpful. To this end, we apply the proposed bias attractor to FixMatch. It can be seen from Table 5 that

Table 5: Ablation study.

| Methods | CIFAR-10 ($\gamma_l = 100$) | |
|---|---|---|
| | $\gamma_u = 100$ | 1/100 (reversed) |
| FixMatch | 71.5 / 68.8 | 65.5 / 26.0 |
| FixMatch w/ bias attractor | 73.9 / 70.7 | 66.6 / 44.8 |
| L2AC w/o bi-level training | 78.4 / 76.6 | 79.3 / 78.0 |
| L2AC (ours) | 82.1 / 81.5 | 82.2 / 81.7 |

the bias attractor helps improve the performance to a certain extent, indicating that the bias attractor is effective yet not significant. As analyzed in Section 3.2, there is no prior knowledge for the bias attractor to assimilate the training bias in this plain training manner. **2)** Next, we study how important the role bi-level optimization plays in our method. Instead of using a bi-level learning framework, we disengage the hierarchy structure of our upper-level loss and lower-level loss, and reformulate a single level optimization problem as $\mathcal{L} + \lambda\mathcal{L}^{bal}$. The results in Table 5 show that such a degraded version of L2AC provides substantial performance gain over *FixMatch w/ bias attractor* while still inferior to our L2AC. This demonstrates the effect of the proposed bias adaptive classifier and bi-level learning framework on protecting the linear classifier from the training bias.

## 5 CONCLUSION

In this work, we propose a bias adaptive classifier to deal with the training bias problem in imbalanced SSL tasks. The bias adaptive classifier is consist of a linear classifier to predict unbiased labels and a bias attractor to assimilate the complicated training bias. It is learned with a bi-level optimization framework. With such a tailored classifier, the unlabeled data can be fully used by online pseudo-labeling to improve the performance of pseudo-labeling SSL methods. Extensive experiments show that our proposed method achieves consistent improvements over the baselines and current state-of-the-arts. We believe that our bias adaptive classifier can also be used for more complex data bias other than class imbalance.

ACKNOWLEDGMENTS

We thank the anonymous reviewers for their constructive suggestions on improving this paper. This research was supported by National Key R&D Program of China (2020YFA0713900), the Macao Science and Technology Development Fund under Grant 0612020A2, The Major Key Project of PCL (PCL2021A12), the China NSFC projects under contract 61721002 and 61906144.

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

## A  ALGORITHM

We give the training algorithm of the proposed L2AC method in Algorithm 1.

---

**Algorithm 1** learning to adapt classifier during training

---

**Input:** labeled / unlabeled training data $\mathcal{D}_l$ / $\mathcal{D}_u$, labeled / unlabeled batch size $n$ / $m$, max iterations $T$

**Output:** classification network parameters $\{\theta, \phi\}$

1: Initialize $\{\theta^0, \phi^0\} \leftarrow \{\theta, \phi\}$ and $\omega^0 \leftarrow \omega$.
2: **for** $t = 0$ **to** $T$ **do**
3:   $\hat{\mathcal{D}}_l = \{\mathbf{x}_i, \mathbf{y}_i\}_{i=1}^n \leftarrow$ SampleMiniBatch$(\mathcal{D}_l, n)$.
4:   $\hat{\mathcal{D}}_u = \{\mathbf{x}_i\}_{i=1}^m \leftarrow$ SampleMiniBatch$(\mathcal{D}_u, m)$.
5:   $\mathcal{B} = \{\mathbf{x}_i, \mathbf{y}_i\}_{i=1}^n \leftarrow$ SampleMiniBatch$(\mathcal{D}_l, n)$.
6:   Estimate pseudo-label $\hat{\mathbf{y}}_i$ for $\mathbf{x}_i \in \hat{\mathcal{D}}_u$.
7:   Compute lower-level loss $\mathcal{L}$ by Eq. (3).
8:   Update network parameters $\{\theta^{t+1}, \phi^{t+1}\}$ by Eq. (4).
9:   Compute upper-level loss $\mathcal{L}^{bal}$ by Eq. (5).
10:   Update bias attractor parameters $\omega^{t+1}$ by Eq. (6).
11: **end for**

---

## B  PROOF OF PROPOSITION 3.1

**Proof B.1** *The update of $\phi$ is as:*

$$\phi^{t+1} = \phi^t - \alpha \frac{1}{n} \sum_{i=1}^n \frac{\partial \mathcal{L}_i(\theta, \phi, w)}{\partial \phi} |_{\phi^t}. \tag{8}$$

*In consequence, we have*

$$
\begin{aligned}
\omega^{t+1} &= \omega^t - \eta \nabla \mathcal{L}^{bal}(\theta, \phi^{t+1})|_{\omega^t} \\
&= \omega^t + \eta \alpha \frac{1}{n} \sum_{i=1}^n \frac{\partial^2 \mathcal{L}_i(\theta, \phi)}{\partial \phi \partial \omega}^T |_{\phi^t, \omega^t} \frac{\partial \mathcal{L}^{bal}(\theta, \phi)}{\partial \phi}|_{\phi^{t+1}} \\
&= \omega^t + \eta \alpha \frac{1}{n} \sum_{i=1}^n \frac{\partial \Delta f_i}{\partial \omega}|_{\omega^t} \left( \frac{\partial^2 \mathcal{L}_i(\theta, \phi)}{\partial \phi \partial \Delta f_i}^T |_{\phi^t} \frac{\partial \mathcal{L}^{bal}(\theta, \phi)}{\partial \phi}|_{\phi^{t+1}} \right),
\end{aligned} \tag{9}
$$

*Denote by $\Xi_i = \frac{\partial \mathcal{L}_i(\theta, \phi)}{\partial \Delta f_i}$, then Eq. (9) becomes*

$$\omega^{t+1} = \omega^t + \eta \alpha \frac{1}{n} \sum_{i=1}^n G_i \frac{\partial \Delta f_i}{\partial \omega}|_{\omega^t}, \tag{10}$$

*where*

$$G_i = \left\langle \frac{\partial \Xi_i}{\partial \phi}|_{\phi^t}, \frac{\partial \mathcal{L}^{bal}(\theta, \phi)}{\partial \phi}|_{\phi^{t+1}} \right\rangle.$$

*As the training loss is*

$$\mathcal{L}_i(\theta, \phi) = \log \sum_{k=1}^d e^{z_{i,k} + \Delta f_{i,k}} - z_{i,c_i} - \Delta f_{i,c_i}, \tag{11}$$

*where $c_i$ is the class label of the $i$-th sample. Therefore*

$$\Xi_{i,k} = \begin{cases} \dfrac{e^{z_{i,k} + \Delta_{i,k}}}{\sum_{s=1}^d e^{z_{i,s} + \Delta_{i,s}}}, & k \neq c_i \\ \dfrac{e^{z_{i,k} + \Delta_{i,k}}}{\sum_{s=1}^d e^{z_{i,s} + \Delta_{i,s}}} - 1, & k = c_i \end{cases} = \mathbf{p}_i - \mathbf{y}_i \tag{12}$$

*Meanwhile, the upper level loss is defined as*

$$\mathcal{L}^{bal}(\theta, \phi) = -\frac{1}{m} \sum_{j=1}^m \mathcal{L}_i^{bal}(\theta, \phi), \tag{13}$$

*this finishes the proof.*

## C  COMPARISON METHODS

To comprehensively evaluate the proposed method, we compare it with the **Vanilla** model that merely trained with labeled data and three other lines of methods: 1) re-balancing methods where only the class-imbalanced labeled data are used for training, including: Re-sampling (Japkowicz, 2000), LDAM-DRW (Cao et al., 2019) and cRT (Kang et al., 2020). 2) pseudo-labeling based SSL methods where both labeled and unlabeled data is used (without considering class-imbalance), including: Pseudo-labels (Lee et al., 2013), MixMatch (Berthelot et al., 2019b) and FixMatch (Sohn et al., 2020). 3) imbalanced semi-supervised learning methods that consider class-imbalance and unlabeled data simultaneously, including: DARP (Kim et al., 2020a), CReST+ (Wei et al., 2021), ABC (Lee et al., 2021), SaR (Lai et al., 2022) and DASO (Oh et al., 2022).

We herein give a brief introduction for all the comparison methods.

- **Vanilla**, a plain classification network, e.g., Wide ResNet-28-2 (Oliver et al., 2018), trained with imbalanced labeled data by cross-entropy loss.
- **Re-sampling**, a re-balancing method that uses re-sampling strategy to balance the distribution of training data.
- **LDAM-DRW**, i.e., Label-distribution-aware margin, a re-weighting method where the classifier encourage to maintain large margin for tail classes.
- **cRT**, i.e., Classifier re-training, a two-stage training method that first pretrains the entire network with all the imbalanced training data and re-train the classifier with a balanced objective.
- **MixMatch**, a SSL method which combines pseudo-labeling and consistency regularization techniques via Mixup augmentation (Zhang et al., 2018).
- **FixMatch**, a pseudo-labelling based SSL method of which the strongly augmented unlabeled samples (whose pseudo labels are generated from their weakly augmented versions) are used to train the network.
- **DARP**, a recent state-of-the-art imbalanced SSL method that refines raw pseudo-labels via a convex optimization for alleviating distribution bias arisen by imbalanced and unlabeled training data.
- **CReST**, a pseudo-labeling based imbalanced SSL method that combines re-balancing and distribution alignment techniques to alleviate the training bias. The method assumes that labeled and unlabeled data have roughly the same distribution.
- **ABC**, which equips with two parallel linear classifiers with one fitting the imbalanced data and the other fitting the re-balanced data, and adds the consistency regularization to further improve the performance.
- **SaR**, i.e., self-adaptive refinement, which proposes the concept of mitigating vector that refines the soft labels of unlabeled data before generating the one-hot pseudo labels to alleviate the confirmation bias brought about by unlabeled samples.
- **DASO**, for an unlabeled sample, which combines its pseudo-label from the linear classifier with that from a similarity-based classifier to leverage their complementary properties in terms of bias. Moreover, a semantic alignment loss is proposed to balance the biased feature representation.

For fair comparison, we use the same code base [3] as DARP (Kim et al., 2020a). As the training or evaluation protocols of CReST (Wei et al., 2021), ABC (Lee et al., 2021) and DASO (Oh et al., 2022) are different from that of DARP, we reproduce their results according to the official codes (i.e., CReST [4], ABC [5] and DASO [6]) released by the authors. Note that the results on CIFAR-100 in DARP (Kim et al., 2020a) are achieved under $N_1 = 300, M_1 = 150$, while this paper keeps $N_1 = 150, M_1 = 300$ for satisfying the common assumption that the amount of unlabeled data are larger than that of labeled data.

---

[3]https://github.com/bbuing9/DARP

[4]https://github.com/google-research/crest

[5]https://github.com/LeeHyuck/ABC

[6]https://github.com/ytaek-oh/daso

# D EXPERIMENTAL SETUPS

## D.1 IMPLEMENTATION DETAILS ON CIFAR AND STL-10

All our experiments are implemented with the Pytorch platform (Paszke et al., 2019) and follows the experimental settings in Kim et al. (2020a). We use Wide ResNet-28-2 (Oliver et al., 2018) as our backbone network. During training, the model is trained with Adam optimizer (Kingma & Ba, 2015) under the default parameter setting, i.e., $\beta_1 = 0.9$, $\beta_2 = 0.999$, and $\epsilon = 10^{-8}$. The learning rate is set as $2 \times 10^{-3}$ and the batch size is set as 64. The total number of training iterations are $2.5 \times 10^5$ as in Kim et al. (2020a). To evaluate the model, we follow the setting in Berthelot et al. (2019b) and use an exponential moving average (EMA) of its parameters with a decay rate of 0.999 at each iteration. We also follow the standard evaluation protocols in Berthelot et al. (2019b) that evaluates the performance at every 500 iterations and reports the average test accuracy of the last 20 evaluations.

**Bias attractor:** As aforementioned in Section 3.2, the bias attractor is a light-weight network, i.e., a multi-layer perceptron with one hidden layer in this paper. The input of the bias attractor is the prediction scores output by the linear classifier, so the input dimension is the same as the number of classes. We normalize the input through its $L_2$ norm or a softmax activation. Concretely, We use softmax operator on CIFAR-10 and STL-10, and $L_2$ norm on CIFAR-100 and SUN-397 due to its better and more stable performance than softmax operator. Note that the gradients of the input of the bias attractor are stopped in the training stage. The hidden layer dimension of the bias attractor is fixed as 256, which keeps stable and sound results through all our experiments. The parameters of bias attractor are updated by Eq. (6), where the learning rate $\eta$ is set as $1 \times 10^{-4}$.

**Baselines:** We evaluate our L2AC based on two recent popular SSL methods, i.e., MixMatch (Berthelot et al., 2019b) and FixMatch (Sohn et al., 2020). Both the two baselines are the cornerstone of current state-of-the-art imbalanced SSL methods, such as DARP (Kim et al., 2020a), CReST (Wei et al., 2021) and DASO (Oh et al., 2022). Note that the two baselines can be uniformly formulated as Eq. (1). For MixMatch, the pseudo-label of one unlabeled example is produced by temperature sharpening to the the average prediction of its different augmented versions, and the objective function of unlabeled data is adopted as mean-squared loss (MSE) function. The threshold $\tau$ is kept as 0, and $\lambda_u$ is dynamically updated by a linear ramp-up strategy, i.e., $\lambda_u$ linearly increases from 0 to 75 during training. For FixMatch, unlabeled data are augmented by weak and strong augmentations via RandAugment (Cubuk et al., 2020). In particular, the weekly augmented data are used to generate pseudo-labels for the strongly augmented data, and these strongly augmented data are then used to compute the unlabeled loss in Eq. (1). We set $\tau$ as 0.95, and $\lambda_u$ as 1 without applying linear ramp-up strategy.

## D.2 IMPLEMENTATION DETAILS ON SUN397

SUN397 (Xiao et al., 2010) is an imbalanced real-world scene classification dataset, which originally consists of 108,754 RGB images labeled with 397 classes. Following the experimental setups in Kim et al. (2020a), we hold-out 50 samples per each class for testing because no official data split is provided. We then artificially construct the labeled and unlabeled dataset using the remaining dataset according to $\frac{M_k}{N_k} = 2$. The comparison methods includes: Vanilla, classifier retraining (cRT) (Kang et al., 2020), FixMatch (Sohn et al., 2020), DARP (Kim et al., 2020a) and ABC (Lee et al., 2021).

**Training details.** For pre-processing, we randomly crop and rescale to $224 \times 224$ size all labeled and unlabeled training images before applying augmentation. We use standard ResNet-34 (He et al., 2016) as our backbone network. During training, the model is trained with Adam optimizer (Kingma & Ba, 2015) with a batch-size of 128 labeled samples and 256 unlabeled samples, and a initial learning rate of 0.002 for 300 training epochs. For fair comparison, all the experiments are based on FixMatch (Sohn et al., 2020). We set unlabeled loss weight $\lambda_u$ as 1.0 and confidence threshold $\tau$ as 0.6, and utilize exponential moving average technique with decay rate 0.99. Following Kim et al. (2020a), we adopt RandAugment with random magnitude (Cubuk et al., 2020) for strong augmentation and random horizontal flip for weak augmentation.

# E  ADDITIONAL EXPERIMENTS

## E.1  EVALUATION ON IMBALANCED TEST SETS

Real-world test data is not always following an uniform distribution, and we herein simulate several imbalanced test sets to further study the generalization of the proposed method. As shown in Fig. 6, we construct three imbalanced test sets: (a) Test-1, which follows a long-tailed class distribution with an imbalance ratio of 10; (b) Test-2, which follows a reversed long-tailed class distribution; (c) Test-3, which follows a random class distribution. We evaluate the proposed L2AC upon FixMatch (Sohn et al., 2020) and compare it with the following methods: DARP (Kim et al., 2020a), CReST+ (Wei et al., 2021), ABC (Lee et al., 2021). All these models are trained on CIFAR-10 with imbalanced ratio $\gamma = \gamma_l = \gamma_u = 100$. As the evaluation metrics bACC and GM are insensitive to class imbalance of test sets, we add ACC to measure the recognition accuracy for all samples.

The results are summarized in Tab. 6. It can be observed that the network learned with our L2AC achieves the best or the second best performance across all the test settings, which further indicates that the proposed bi-level learning framework provides a relatively unbiased classifier compared to other comparison methods.

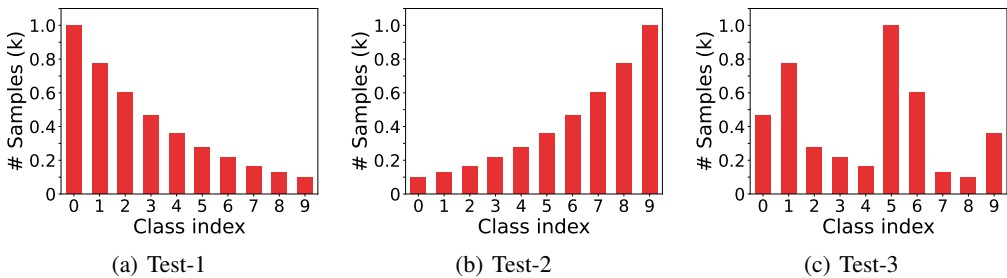

|     (a) Test-1     |     (b) Test-2     |     (c) Test-3     |

Figure 6: Class distributions of three typical imbalanced test sets.

Table 6: Imbalanced test set results. ACC: accuracy for all samples.

| Methods | Test-1 | | | Test-2 | | | Test-3 | | |
|---|---|---|---|---|---|---|---|---|---|
|  | bACC | GM | ACC | bACC | GM | ACC | bACC | GM | ACC |
| FixMatch (Sohn et al., 2020) | 72.4 | 66.3 | 86.1 | 72.7 | 66.9 | 56.6 | 72.3 | 65.7 | 75.6 |
| w/ DARP (Kim et al., 2020a) | 74.8 | 72.5 | 86.3 | 75.5 | 73.2 | 63.9 | 75.6 | 73.3 | 77.4 |
| w/ CReST (Wei et al., 2021) | 77.8 | 76.5 | 86.3 | 77.2 | 74.8 | 68.9 | 77.5 | 76.3 | 80.3 |
| w/ ABC (Lee et al., 2021) | 80.2 | 79.2 | **88.1** | 80.2 | 79.0 | 71.7 | 80.1 | 79.0 | 82.8 |
| w/ L2AC (ours) | **82.6** | **82.0** | 87.2 | **82.4** | **81.8** | **78.6** | **82.7** | **82.1** | **83.9** |

## E.2  EVALUATION ON STEP IMBALANCE SETUPS

In the main text, we have evaluated the proposed L2AC under long-tailed imbalanced settings with various imbalance ratios. Here, we further validate its generalization in the case of step imbalance on CIFAR-10 under two typical setups, namely $\gamma_l = \gamma_u = 100$ and $\gamma_= 100, \gamma_u = 1$. As shown in Fig. 7, step imbalance assumes a more severely imbalanced class distribution than the long-tailed imbalance setups, as there are very scare data for half of the classes. The experimental results are summarized in Tab. 7. We can see that the proposed L2AC achieved the best performance compared with all the comparison methods for all settings. Especially in the case of distribution mismatch between labeled and unlabeled data, L2AC brings significant performance gain.

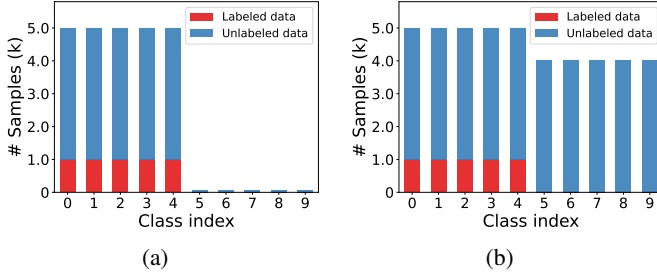

(a)                    (b)

Figure 7: Class distributions of CIFAR-10 under step imbalance setups with (a) $\gamma_l = \gamma_u = 100$; (b) $\gamma_l = 100, \gamma_u = 1$.

Table 7: Performance (bACC / GM) on CIFAR-10 under step imbalance setups.

| Methods | $\gamma_l = \gamma_u = 100$ | $\gamma_l = 100, \gamma_u = 1$ |
|---|---|---|
| FixMatch (Sohn et al., 2020) | $55.0_{\pm 0.84}$ / $24.4_{\pm 2.97}$ | $60.1_{\pm 1.97}$ / $18.32_{\pm 3.16}$ |
| w/ DARP (Kim et al., 2020a) | $58.6_{\pm 0.44}$ / $30.0_{\pm 1.27}$ | - |
| w/ ABC (Lee et al., 2021) | $75.1_{\pm 0.78}$ / $67.5_{\pm 0.97}$ | $75.8_{\pm 0.98}$ / $69.2_{\pm 1.35}$ |
| w/ L2AC (ours) | $\mathbf{76.7}_{\pm 0.41}$ / $\mathbf{74.5}_{\pm 0.61}$ | $\mathbf{81.8}_{\pm 0.87}$ / $\mathbf{80.5}_{\pm 1.01}$ |

### E.3 DETAILED ANALYSIS ON THE QUALITY OF PSEUDO-LABELS.

**Case of $\gamma_l = \gamma_u$.** Fig. 8 visualizes the confusion matrices of pseudo-labels of Fixmatch (Sohn et al., 2020) and our L2AC under the imbalance ratio $\gamma_l = \gamma_u = 100$. Note that this requires to use true labels that are hidden in the training stage. We can observe that the original pseudo-labels are highly biased toward majority classes of the labeled dataset. In contrast, our L2AC tends to a relatively equal per-class recall, especially on minority classes the performance is significantly improved compared to FixMatch (Sohn et al., 2020). This suggests that the proposed method readily improves the quality of pseudo-labels.

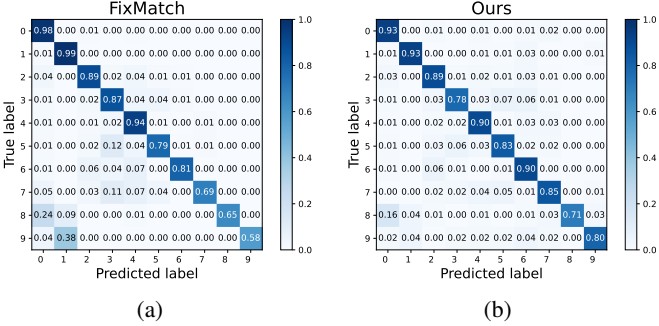

(a)                    (b)

Figure 8: Pseudo-label confusion matrices of (a) FixMatch and (b) Ours on CIFAR-10 under $\gamma_l = \gamma_u = 100$.

**Case of $\gamma_l \neq \gamma_u$.** Fig. 9 visualizes the confusion matrices of pseudo-labels for Fixmatch (Sohn et al., 2020) and our L2AC under the imbalance ratio $\gamma_l = 100$ while $\gamma_u = 100$ (Reversed). Under this setting, labeled set is unchanged and the total number of the training data remains the same compared to $\gamma_l = \gamma_u = 100$. Unlabeled data follow a reversed class distribution, which means that more samples are added to the training set for the minority classes. According to the confusion matrix in Fig. 9, FixMatch (Sohn et al., 2020) remains to achieve high recall on the majority classes and low recall on the minority classes. Most samples of the last two tail classes are mislabeled as the two most majority classes. These findings reveal that the pseudo-labeling process becomes

more biased as the mismatch of the distributions of labeled and unlabeled data become severe. The confusion matrix of L2AC clearly shows that it can exactly assimilate the training bias through the proposed bias attractor such that the linear classifier is able to predict correct label.

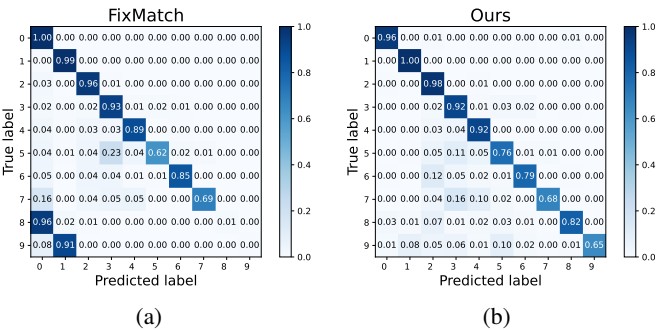

(a)                                    (b)

Figure 9: Pseudo-label confusion matrices of (a) FixMatch and (b) Ours on CIFAR-10 with $\gamma_l = 100, \gamma_u = 100$ (reversed).

### E.4   TRAINING CONVERGENCE VERIFICATION

To verify the convergence of our proposed L2AC approach, Fig. 10 visualizes the training curves of the lower-level loss Eq. (3) and the upper-level loss Eq. (5) with training iteration increasing from 0 to $2.5 \times 10^5$. We can see that both lower-level loss and upper-level loss convergence fast within the first 100 epochs ($5 \times 10^4$ iterations), and the test accuracy curve increases fast at the same time. When the test accuracy reached the peak value, our L2AC roughly remains the same test accuracy until termination, which verifies the robustness of the proposed method.

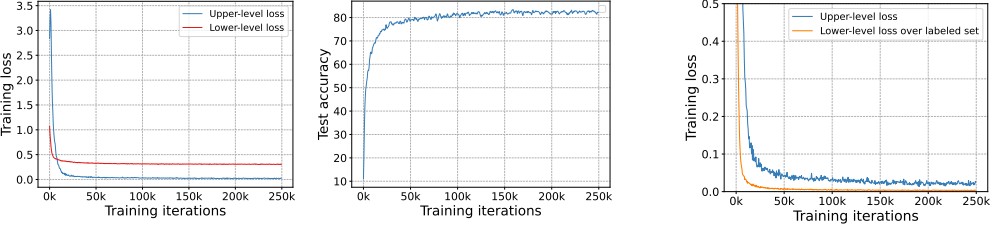

Figure 10: Curves of **Left:** lower-level and upper-level losses and **Right:** test accuracy durning training of our approach on long-tailed CIFAR-10 under $\gamma_l = \gamma_u = 100$.

Figure 11: Curves of upper-level loss (over $\mathcal{B}$) and lower-level loss over labeled set $\mathcal{D}_l$.

Since the balanced set $\mathcal{B}$ is dynamically sampled from the labeled set $\mathcal{D}_l$, a nature question is whether the upper-level loss (over $\mathcal{B}$) has the same convergence rate with the lower-level loss (over $D_l$) during training. To investigate this, we further visualize the curves of these two losses in Fig. 11, and we can observe that the two losses decrease differently and converge to values of different magnitudes at different iterations.

### E.5   RUNNING COST ANALYSIS

In Section 3.2, we conduct a complexity analysis of the training algorithm of our L2AC, which shows that our method is very efficient in theory. To verify this, we herein measure floating point operations per second (FLOPS) using NVIDIA GeForce RTX 3090 to quantify the training cost. We compare our proposed algorithm with the baseline model FixMatch (Sohn et al., 2020) and two state-of-the-art methods (Kim et al., 2020a) and (Lee et al., 2021), as L2AC uses the same code base with

these two methods. Besides, we also provide the training cost of L2AC (traditional), the algorithm that unrolls the gradient of the whole classification network to compute the second-order gradient of the bias attractor, just like most gradient-based bi-level optimization algorithm (Finn et al., 2017; Ren et al., 2018). It can be seen that: (1) our L2AC is much faster than L2AC (traditional); (2) The training cost of our L2AC is comparable to the current state-of-the-art method ABC (Lee et al., 2021).

Table 8: Training cost analysis on CIFAR-10 and CIFAR-100.

| Methods | Params | FLOPS | |
| --- | --- | --- | --- |
| | | CIFAR-10 | CIAFR-100 |
| FixMatch (Sohn et al., 2020) | 1.47 M | 19.6 iter/sec | 19.6 iter/sec |
| w/ DARP (Kim et al., 2020a) | 1.47 M | 18.2 iter/sec | 7.5 iter/sec |
| w/ ABC (Lee et al., 2021) | 1.47 M | 15.1 iter/sec | 14.9 iter/sec |
| w/ L2AC (traditional) | 1.48 M | 9.9 iter/sec | 9.7 iter/sec |
| w/ L2AC (ours) | 1.48 M | 14.2 iter/sec | 13.9 iter/sec |

It can be observed that the computation cost increment of our proposed L2AC is nearly negligible compared with the baseline models especially considering the significant improvement performance of L2AC. And it is worth noting that our proposed L2AC is more efficient than traditional second-order optimization, we think it is owing to two reasons: (1) the bias attractor only adds a very small number of parameters (about 0.68% of the total number of parameters); (2) To calculate the second-order gradient of these parameters, we only need to unroll the gradient of the linear classifier as shown in Eq. (4).

Note that in the test stage our L2AC requires no extra overhead compared with the baseline model FixMatch.

### E.6 FEATURE VISUALIZATION UNDER OTHER SETUPS

In Fig. 5, we visualize the representations of training data through t-SNE (Van der Maaten & Hinton, 2008) on CIFAR-10 with $\gamma_l = 100, \gamma_u = 1$. Under such a setting, each class roughly has the same number of samples, which ensures a good visual visualization for each class. To verify that the proposed L2AC can generally obtain a high-quality representation, we further visualize the t-SNE of training data under other experimental setups, including $\gamma_l = \gamma_u = 100$ and $\gamma_l = 100, \gamma_u = 100$ (reversed). As shown in Fig. 12 and Fig. 13, compared with FixMatch (Sohn et al., 2020), our L2AC certainly improves the separability of the tail classes from the head classes. This verifies that the result in Fig. 5 is not owing to the specific choice of the experimental setups.

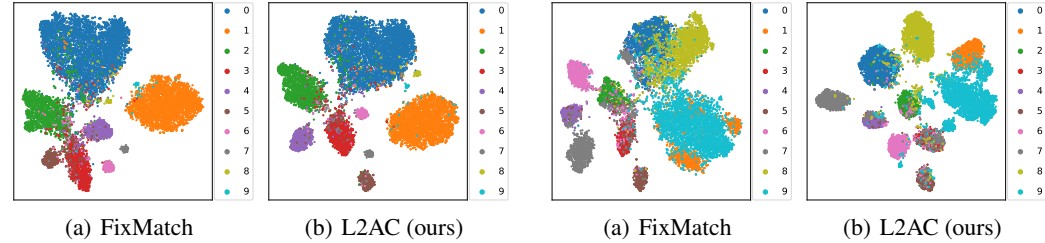

(a) FixMatch      (b) L2AC (ours)      (a) FixMatch      (b) L2AC (ours)

Figure 12: t-SNE visualization of unlabeled data for (a) FixMatch and (b) L2AC on CIFAR-10 with $\gamma_l = \gamma_u = 100$.

Figure 13: t-SNE visualization of unlabeled data for (a) FixMatch and (b) L2AC on CIFAR-10 with $\gamma_l = 100, \gamma_u = 100$ (reversed).

# F  CONVERGENCE ANALYSIS OF ALGORITHM 1

We prove that our algorithm to minimize $\mathcal{L}^{bal}$ converges at rate of $\tilde{\mathcal{O}}(\frac{1}{\sqrt{T}})$, which meets the convergence results of similar work such as Ren et al. (2018).

**Theorem F.1** *Assume that $\mathcal{L}^{bal}$ in Eq. (5) is $L$-smooth with $\rho$-bounded gradients on the samples of balanced set $\mathcal{B}$, and the bias attractor function $\Delta f_w(\cdot)$ is differentiable with a $\delta$-bounded gradient and twice differentiable with bounded Hessian. Let $\alpha_t$ in Eq. (4) satisfies $\alpha_t = \frac{c_1}{t} \leq \frac{2}{L}$, and $\eta_t$ in Eq. (6) is set as $\eta_t = \frac{c_2}{\sigma\sqrt{t}}$ for $c > 0$ such that $\frac{c}{\sigma\sqrt{t}} < \frac{1}{L}$. Then the loss function $\mathcal{L}^{bal}$ converges to critical point at the rate $\tilde{\mathcal{O}}(\frac{1}{\sqrt{T}})$ as*

$$\min_{0 \leq t \leq T} \mathbb{E}[\|\nabla \mathcal{L}^{bal}(\phi^t(\omega^t))\|_2^2] \leq \tilde{\mathcal{O}}(\frac{1}{\sqrt{T}}). \tag{14}$$

**Proof F.1** *The parameter $\omega$ is updated by stochastic gradient descent as*

$$\omega^{t+1} = \omega^t - \eta_t \nabla \mathcal{L}^{bal}(\phi^t(\omega^t))|_{\mathcal{B}_t}, \tag{15}$$

*where $\mathcal{B}_t$ is a mini-batch of class-balanced data. We can write Eq. (15) as*

$$\omega^{t+1} = \omega^t - \eta_t [\nabla \mathcal{L}^{bal}(\phi^t(\omega^t)) + \xi^t], \tag{16}$$

*where $\xi^t$ is the random gradient noise with zero mean and finite variance $\sigma$. Observe that*

$$\begin{aligned}
&\mathcal{L}^{bal}(\phi^{t+1}(\omega^{t+1})) - \mathcal{L}^{bal}(\phi^t(\omega^t)) \\
&= \{\mathcal{L}^{bal}(\phi^{t+1}(\omega^{t+1})) - \mathcal{L}^{bal}(\phi^t(\omega^{t+1}))\} + \{\mathcal{L}^{bal}(\phi^t(\omega^{t+1})) - \mathcal{L}^{bal}(\phi^t(\omega^t))\}.
\end{aligned} \tag{17}$$

*According to $L$-smoothness of $\mathcal{L}^{bal}$ with respect to $\phi$, we have*

$$\begin{aligned}
&\mathcal{L}^{bal}(\phi^{t+1}(\omega^{t+1})) - \mathcal{L}^{bal}(\phi^t(\omega^{t+1})) \\
&\leq \langle \nabla_\phi \mathcal{L}^{bal}(\phi^t(\omega^{t+1})), \phi^{t+1}(\omega^{t+1}) - \phi^t(\omega^{t+1}) \rangle + \frac{L}{2}\|\phi^{t+1}(\omega^{t+1}) - \phi^t(\omega^{t+1})\|_2^2,
\end{aligned} \tag{18}$$

*meanwhile*

$$\phi^{t+1}(\omega^{t+1}) - \phi^t(\omega^{t+1}) = -\alpha_t \nabla_\phi \mathcal{L}(\phi(\omega^{t+1}), \theta)|_{\phi^t(\omega^{t+1})}, \tag{19}$$

*we have*

$$\begin{aligned}
&\mathcal{L}^{bal}(\phi^{t+1}(\omega^{t+1})) - \mathcal{L}^{bal}(\phi^t(\omega^{t+1})) \\
&\leq \|\nabla_\phi \mathcal{L}^{bal}(\phi^t(\omega^{t+1}))\| \| - \alpha_t \nabla_\phi \mathcal{L}(\phi(\omega^{t+1}), \theta)|_{\phi^t(\omega^{t+1})}\| + \frac{L}{2}\| - \alpha_t \nabla_\phi \mathcal{L}(\phi(\omega^{t+1}), \theta)|_{\phi^t(\omega^{t+1})}\|_2^2, \\
&\leq \alpha_t \rho^2 + \frac{L}{2}\alpha_t^2 \rho^2,
\end{aligned} \tag{20}$$

*the second inequality holds due to the assumption $\|\nabla_\phi \mathcal{L}^{bal}(\phi^t(\omega^{t+1}))\| \leq \rho$ and $\| - \alpha_t \nabla_\phi \mathcal{L}(\phi(\omega^{t+1}), \theta)|_{\phi^t(\omega^{t+1})}\| \leq \rho$.*

*According to the $L$-smoothness of $\mathcal{L}^{bal}$ with respect to $\omega$, we have*

$$\begin{aligned}
&\mathcal{L}^{bal}(\phi^t(\omega^{t+1})) - \mathcal{L}^{bal}(\phi^t(\omega^t)) \\
&\leq \langle \nabla_\omega \mathcal{L}^{bal}(\phi^t(\omega^t)), \omega^{t+1} - \omega^t \rangle + \frac{L}{2}\|\omega^{t+1} - \omega^t\|_2^2 \\
&= \langle \nabla_\omega \mathcal{L}^{bal}(\phi^t(\omega^t)), -\eta_t[\nabla_\omega \mathcal{L}^{bal}(\phi^t(\omega^t)) + \xi^t]\rangle + \frac{L\eta_t^2}{2}\|\nabla_\omega \mathcal{L}^{bal}(\phi^t(\omega^t)) + \xi^t\|_2^2 \\
&= (\frac{L\eta_t^2}{2} - \eta_t)\|\nabla_\omega \mathcal{L}^{bal}(\phi^t(\omega^t))\|_2^2 + \frac{L\eta_t^2}{2}\|\xi^t\|_2^2 + (L\eta_t^2 - \eta_t)\langle \nabla_\omega \mathcal{L}^{bal}(\phi^t(\omega^t)), \xi^t\rangle.
\end{aligned} \tag{21}$$

*Then Eq. (17) becomes*

$$\begin{aligned}
&\mathcal{L}^{bal}(\phi^{t+1}(\omega^{t+1})) - \mathcal{L}^{bal}(\phi^t(\omega^t)) \\
&\leq \alpha_t \rho^2 + \frac{L}{2}\alpha_t^2 \rho^2 + (\frac{L\eta_t^2}{2} - \eta_t)\|\nabla_\omega \mathcal{L}^{bal}(\phi^t(\omega^t))\|_2^2 + \frac{L\eta_t^2}{2}\|\xi^t\|_2^2 + (L\eta_t^2 - \eta_t)\langle \nabla_\omega \mathcal{L}^{bal}(\phi^t(\omega^t)), \xi^t\rangle.
\end{aligned} \tag{22}$$

*Rearranging Eq. (22), we have*

$$(\eta_t - \frac{L\eta_t^2}{2})\|\nabla_\omega \mathcal{L}^{bal}(\phi^t(\omega^t))\|_2^2$$

$$\leq \mathcal{L}^{bal}(\phi^t(\omega^t)) - \mathcal{L}^{bal}(\phi^{t+1}(\omega^{t+1})) + \alpha_t\rho^2 + \frac{L}{2}\alpha_t^2\rho^2 + \frac{L\eta_t^2}{2}\|\xi^t\|_2^2 + (L\eta_t^2 - \eta_t)\langle \nabla_\omega \mathcal{L}^{bal}(\phi^t(\omega^t)), \xi^t]\rangle. \tag{23}$$

*Summing up the above inequalities, we obtain*

$$\sum_{t=1}^{T}(\eta_t - \frac{L\eta_t^2}{2})\|\nabla_\omega \mathcal{L}^{bal}(\phi^t(\omega^t))\|_2^2$$

$$\leq \mathcal{L}^{bal}(\phi^1(\omega^1)) - \mathcal{L}^{bal}(\phi^{T+1}(\omega^{T+1})) + \sum_{t=1}^{T}\alpha_t\rho^2(1 + \frac{L}{2}\alpha_t) + \sum_{t=1}^{T}\frac{L\eta_t^2}{2}\|\xi^t\|_2^2 \tag{24}$$

$$+ \sum_{t=1}^{T}(L\eta_t^2 - \eta_t)\langle \nabla_\omega \mathcal{L}^{bal}(\phi^t(\omega^t)), \xi^t]\rangle.$$

$$\leq \mathcal{L}^{bal}(\phi^1(\omega^1)) + \sum_{t=1}^{T}\alpha_t\rho^2(1 + \frac{L}{2}\alpha_t) + \sum_{t=1}^{T}\frac{L\eta_t^2}{2}\|\xi^t\|_2^2 + \sum_{t=1}^{T}(L\eta_t^2 - \eta_t)\langle \nabla_\omega \mathcal{L}^{bal}(\phi^t(\omega^t)), \xi^t]\rangle.$$

*Taking expectations with respect to $\xi^t$ on Eq. (24), we can obtain*

$$\sum_{t=1}^{T}(\eta_t - \frac{L\eta_t^2}{2})\mathbb{E}\|\nabla_\omega \mathcal{L}^{bal}(\phi^t(\omega^t))\|_2^2$$

$$\leq \mathcal{L}^{bal}(\phi^1(\omega^1)) + \sum_{t=1}^{T}\alpha_t\rho^2(1 + \frac{L}{2}\alpha_t) + \sum_{t=1}^{T}\frac{L\eta_t^2\sigma^2}{2} + \sum_{t=1}^{T}(L\eta_t^2 - \eta_t)\langle \nabla_\omega \mathcal{L}^{bal}(\phi^t(\omega^t)), \mathbb{E}_{\xi^t}(\xi^t)]\rangle \quad (25)$$

$$= \mathcal{L}^{bal}(\phi^1(\omega^1)) + \sum_{t=1}^{T}\alpha_t\rho^2(1 + \frac{L}{2}\alpha_t) + \sum_{t=1}^{T}\frac{L\eta_t^2\sigma^2}{2}.$$

*The first inequality holds due to $\mathbb{E}[\|\xi^t\|_2^2] = \sigma^2$ and the second equality holds due to $\mathbb{E}_{\xi^t}[\xi^t] = 0$. Further more, we have*

$$\min_t \mathbb{E}[\|\nabla_\omega \mathcal{L}^{bal}(\phi^t(\omega^t))\|_2^2]$$

$$\leq \frac{1}{\sum_{t=1}^{T}(2\eta_t - L\eta_t^2)}\left[2\mathcal{L}^{bal}(\phi^1(\omega^1)) + \sum_{t=1}^{T}\alpha_t\rho^2(2 + L\alpha_t) + \sum_{t=1}^{T}L\eta_t^2\sigma^2\right]$$

$$\leq \frac{1}{\sum_{t=1}^{T}\eta_t}\left[2\mathcal{L}^{bal}(\phi^1(\omega^1)) + \sum_{t=1}^{T}\alpha_t\rho^2(2 + L\alpha_t) + \sum_{t=1}^{T}L\eta_t^2\sigma^2\right]$$

$$\leq \frac{1}{\sum_{t=1}^{T}\eta_t}\left[2\mathcal{L}^{bal}(\phi^1(\omega^1)) + 4\rho^2\sum_{t=1}^{T}\alpha_t + \sigma^2L\sum_{t=1}^{T}\eta_t^2\right] \tag{26}$$

$$= \frac{1}{\sqrt{T}}\left[2\mathcal{L}^{bal}(\phi^1(\omega^1)) + 4\rho^2\log(T) + \sigma^2L\log(T)\right]$$

$$= \tilde{\mathcal{O}}(\frac{1}{\sqrt{T}}).$$

*The second inequality hods due to $\sum_{t=1}^{T}(2\eta_t - L\eta_t^2) \geq \sum_{t=1}^{T}\eta_t$. The third inequality holds due to $\alpha_t \leq \frac{2}{L}$. As $\alpha_t = \frac{c_1}{t}$ and $\eta_t = \frac{c_2}{\sqrt{t}}$, we have $\sum_{t=1}^{T}\eta_t = (\sqrt{T})$, $\sum_{t=1}^{T}\eta_t^2 = \log(T)$ and $\sum_{t=1}^{T}\alpha_t = \log(T)$ thus the last equality holds, this finish our proof.*

