# OpenReview forum: "Imbalanced Semi-supervised Learning with Bias Adaptive Classifier"
_ICLR.cc/2023/Conference — ICLR 2023 poster_

### Official Review · Reviewer_5NiX · 2022-10-22

**Confidence:** 4
**Correctness:** 3
**Technical Novelty And Significance:** 3
**Empirical Novelty And Significance:** 3
**Recommendation:** 8

**Clarity, Quality, Novelty And Reproducibility:**

This paper is well-written and organized. It proposes novel solution for class-imbalanced SSL and the proposed method is relatively straightforward. With detailed information on training recipe, I think it would not be difficult to reproduce it.

**Strength And Weaknesses:**

This paper is well written and easy to follow. The idea of using another classifier to rectify bias introduced in the original classifier is intuitively sound. Moreover, the formula and design is actually very straightforward yet effective, allowing others to easily reproduce the proposed method and add the proposed module into any SSL framework. Technically, the approach of bi-level training makes a lot of sense to me, when considering the cost function to be enforcing the similarity between gradient of a sample and the average gradient of a sampled balanced set. In addition, this architecture seems works pretty well under various imbalance conditions.

This work relaxes the assumption used by some existing works that unlabeled data and labeled data should follow similar distribution, and experiments show that it can handle both this case and reversed case (unlabeled data distribution is reversed from labeled data distribution). Apart from consistent gain over compared algorithms, this paper also provides solid ablation study to highlight a few important observations on how the proposed algorithm performs regarding different perspectives, such as pseudo-label quality improvement, capability of handling extreme label ratios. Such study is valuable and helps evaluate the contribution of this work, making this paper convincing.

There are several minor weakness, listed as follows:

- Although the related works section covers most of recent SSL algorithms for class imbalanced problem, there are more papers that should be cited:

   -  [1] Class-Imbalanced Semi-Supervised Learning with Adaptive Thresholding, 2022
   -  [2] Debiased Learning from Naturally Imbalanced Pseudo-Labels, 2022

  In particular, [2] also talks about debasing for pseudo-labels under class imbalance scenario, which is quite related to this paper. It would be good to include and discuss about them.

-  The motivation of the proposed bias adaptive classifier is still a bit vague to me, although the structure looks simple. I understand that it takes into account the second order gradient when computing the upper training loss $L_{bal}$, but could not draw a crystal clear connection between updating bias attractive parameters and enforcing model to make more prediction towards the minority classes. How such architecture, i.e., residual connection, linear and non-linear classifier design, and bi-level training helps alleviate classifier bias? More intuitive explanations would help. Additionally, how does the sampled balanced set affect performance of the bias adaptive classifier?

Some minor grammatical errors:

- Note that CReST+ (Wei et al., 2021) fail in … -> fails in
- In summary, such an nested training… -> such a
- design a be-level learning algorithm … -> a bi-level

**Summary Of The Paper:**

This paper proposes a semi-supervised learning algorithm for unbalanced data. The key component is a bias adaptive classifier to alleviate the problem that the model is usually biased towards the majority classes. The bias adaptive classifier, together with the regular classifier, are learned through a bi-level training protocol, where the model us updated first and a sampled balanced set is used to update the bias attractor parameters. Experiments on several long-tailed dataset show the outstanding performance compared to existing solutions.


**Summary Of The Review:**

In summary, this is overall a well-written, well-organized paper and the proposed method is technically sound. Despite its simplicity, the proposed method consistently outperforms existing works on SSL for class imbalance problem, and the improvement is quite significant. There are a few unclear pieces regarding motivation of the bias attractor parameters and missing references.

---

> ### Author Response · Authors · 2022-11-18
> **Response to Reviewer 5NiX**
>
> Thanks a lot for reviewing our paper and acknowledging our contributions. We provide detailed responses below and please kindly let us know if your concerns are addressed.
>
> #### **Q1: There are more papers [1,2] should be cited and discussed.**
>
> **R1:** Thank you for the suggestion. We have cited and discussed the two methods in the related work section of the revised paper.
> > ...Guo & Li (2022) found a fixed threhold for pseudo-labeled sample selection biased towards head classes and in turn proposed to optimize an adaptive threhold for each class.
>
> As well as:
> > ...Most recently, Wang et al. (2022) proposed to combine counterfactual reasoning and adaptive margins to remove the bias from the pseudo-labels. Our approach differs from this method in both model formulation and loss function.
>
> To further compare with DebiasPL [2], we also evaluate our L2AC under DebiasPL’s protocol and report the results in the following table. We hope this help to answer your concern.
>
> **Table R-4:** Results (bACC) on CIFAR-10 under different settings, where $\gamma$ and $\beta$ denote class imbalance ratio and labeled data percentage, respectively. The results of DebiasPL are copied from the original paper.
> | Method | $\gamma=100,\beta=10%\quad$ | $\gamma=100, \beta=30%\quad$ |
> | :---- | :----: | :---: |
> | DebiasPL | $79.2_{\pm1.0}$ | $80.6_{\pm0.5}$ |
> | L2AC (ours)| $79.3_{\pm0.6}$ | $83.6_{\pm0.2}$|
>
> - - -
> #### **Q2: What is the connection between updating bias attractive parameters and enforcing the model to make more predictions towards the minority classes?**
>
> **R2:** Thanks for your valuable comment. On the one hand, the upper-level loss encourages the linear classifier to have basically equal preference for each class (via the empirical loss minimization over a sampled class-balanced set), which alleviates the imbalance bias and helps improve the performance of the minority classes; On the other hand, in the upper-level loss, the linear classifier is an implicit function of the bias attractor parameters, so the updating of the bias attractor aims to obtain a better linear classifier that can minimize the upper-level loss.
>
> - - -
> #### **Q3: How such architecture, i.e., residual connection, linear and non-linear classifier design, and bi-level training helps alleviate classifier bias?**
>
> **R3:** Thanks for your valuable comment. We herein give a brief explanation of what each module is designed for:
> - The linear classifier is the original component of the classification network, and it easily overfits the training bias in imbalanced SSL tasks. This paper targets to protect the linear classifier from the training bias during training, so as NOT to change the original classification network during the test stage.
> - The non-linear bias attractor aims to assimilate complicated training bias, because a non-linear neural network has a powerful expressive ability to approximate almost any continuous function in theory.
> - The residual connection bridges the linear classifier and the bias attractor, which makes the bias attractor a plug-in module, i.e., assimilates the training bias during training and be removed in the test stage. Also, the residual connection has been proven to be helpful for the backpropagation of deep networks.
> - With the bias attractor, we can only *theoretically* fit the complicated training bias, but how to make the bias attractor assimilate the training bias practically (in other words, how to make the linear classifier exactly fit the unbiased class-conditional distribution) is what the bi-level training does. Specifically, the lower-level loss aims to fit the biased training data for representation learning, just like most pseudo-labeling methods do, and the upper-level one targets learning the bias attractor through which the training bias can be decoupled from the linear classifier.
>
> #### **Q4: How does the sampled balanced set affect the performance of the bias adaptive classifier?**
>
> **R4:** The sampled balanced set used in the upper-level optimization problem is unbiased (clean and class-balanced), which determines what kind of class-conditional distribution the linear classifier will fit.
>
> #### **Q5: Some minor grammatical errors.**
>
> **R5:** Many thanks for the careful check of details. We have fixed these errors in our revised version.
>
> #### **Reference**
> >[1] Guo LZ and Li YF. Class-imbalanced semi-supervised learning with adaptive thresholding. In ICML, 2022.
> [2] Wang X, Wu Z, et al. Debiased learning from naturally imbalanced pseudo-labels. In CVPR, 2022.

---

> > ### Comment · Reviewer_5NiX · 2022-12-05
> > **Keeping my rating**
> >
> > Appreciate the feedback from the authors. The response has largely addressed my concerns and I would like to see them in the final version of the paper. I think it's a good paper so I'll keep my rating.

---

> > > ### Author Response · Authors · 2022-12-05
> > > **Thanks for your response**
> > >
> > > Thank you for your additional efforts and time to carefully read our response. We are glad to hear that our response successfully addressed your concerns. Thank you for your valuable comments that helped us improve our paper!

---

> ### Author Response · Authors · 2022-12-03
> **Looking forward to your reply and the discussions**
>
> Dear reviewer 5NiX,
>
> We sincerely thank you for your efforts in reviewing our paper. Since only a few days remain in the discussion period, please let us know if our responses have well addressed your concerns. If you have any further questions, we are more than happy to address them. Thanks again for your valuable suggestions!
>
> Best regards,
>
> All anonymous authors

---

### Official Review · Reviewer_ihj1 · 2022-10-23

**Confidence:** 4
**Correctness:** 3
**Technical Novelty And Significance:** 3
**Empirical Novelty And Significance:** 3
**Recommendation:** 8

**Clarity, Quality, Novelty And Reproducibility:**

The explanation of the proposed method could be improved. Despite the novelty and quality of the proposed method, the explanation in Section 3.2 does not provide clear intuition why the proposed method works and the idea is effective. The current explanation says that a residual connection is key. However, the explanation, such as "the proposed bias attractor can assimilate a wide range of training biases", does not provide an in-depth understanding of why the proposed method works. My current understanding is that $\Delta f_\omega$ absorbs the bias from imbalanced distributions. I would be grateful if more in-depth explanations were provided for the proposed method.

Some notations are ambiguous. In Eq.(2), $\boldsymbol{I}$ and $\circ$ seem not to be defined. Also, $|_{\phi^t}^T$ in $G_i$ in Proposition 3.1.

In Figure 5, $\gamma_l=100$ and $\gamma_u=1$ were chosen for t-SNE visualization. It is interesting to see the results of the other values of $\gamma_l$ and $\gamma_u$.
Such a result might show that a high-quality representation can often be obtained by the proposed method, i.e., the result in Figure 5 is not owing to the specific choice of the hyperparameters.

Appendices B and C seems to be useful information for reproducibility. Also, the authors declare that the source code will be made publicly available in Section 1. The results in this paper might be able to be reproducible by the code.

**Strength And Weaknesses:**

##### Strengths
- The idea of the proposed bias adaptive classifier is novel and has an impact on imbalanced SSL.
- The bi-level training is important to train a bias adaptive classifier and its effectiveness is carefully investigated through an ablation study.
- The superior performance of the proposed method was demonstrated on extensive numerical experiments and various recent SSL methods were compared with the proposed method.

##### Weaknesses
- The current explanation of the proposed method is not really clear. The presentation of the proposed method could be improved.
- It seems that the proposed method requires intensive computation compared with the other methods because the proposed method requires a second-order gradient. If the computation time is provided during the rebuttal phase, I will reconsider this point.

**Summary Of The Paper:**

This paper proposed a semi-supervised learning (SSL) method, especially for imbalanced classification. Since standard SSL methods based on pseudo labels often assign labels of head classes to unlabeled data, the obtained pseudo labels tend to have a small number of labels for tail classes, resulting in low classification performance. To cope with the problem, the bias adaptive classifier is proposed to reduce bias from imbalanced data. To train neural networks with the bias adaptive classifier, a bi-level optimization approach is employed. Through extensive numerical experiments, the proposed method outperformed various recent SSL methods. Also, the ablation study showed the usefulness of the proposed bi-level training with the bias adaptive classifier.

**Summary Of The Review:**

This paper proposed the bias adaptive classifier for imbalanced semi-supervised learning. The idea looks novel, and the empirical results support its effectiveness. The reproducibility seems high. If the clarity of this paper is improved, the overall score will be increased.

---

> ### Author Response · Authors · 2022-11-18
> **Response to Reviewer ihj1 (Part 1)**
>
> Many thanks for your review and constructive suggestions to improve the paper. We respond to your comments one-by-one in what follows. In the revised paper, we highlighted major revisions in blue. Please kindly let us know if your concerns are addressed and whether you have any further concerns.
>
> #### **Q1: In-depth explanations of the proposed method, especially for the types of training biases and the details of the proposed method.**
> **R1:** Thank you for pointing this out. Actually, we think pseudo-labeling methods mainly suffer from: (1) *Imbalance Bias*, caused by the imbalanced class distribution of the labeled data; (2) *Pseudo-label Bias*, arising from inaccurate pseudo-labels of unlabeled data.
>
> To illustrate these two biases, we revised Fig. 1(b) and (c) by adding the performance of a lower bound model (trained with the imbalanced labeled data only). It is clear that
> - Imbalance Bias exists because the performance of the baseline model FixMatch is positively correlated with the label distribution.
> - Pseudo-label Bias also exists as the performance of FixMatch on tail classes is even worse than the lower bound model, indicating that inaccurate pseudo-labels generated in the training process further bias the model.
>
> Not that these two types of training bias are tightly coupled in the training process, and that is why we need a bias attractor (a nonlinear network with powerful expressive ability) to fit them. In fact, the bias attractor *only theoretically* ensures that the bias can be assimilated. How to fit the training bias practically (in other words, how to make the linear classifier exactly fit the unbiased class-conditional distribution) is what our bi-level optimization strategy does. Specifically, the lower-level loss fit the biased training data for representation learning (just like most pseudo-labeling methods do), and the upper-level one targets to decouple the training bias from the linear classifier via empirical risk minimization over *any* sampled balanced set $\mathcal B$.
>
> As for the residual connection, it is the key to conveniently make the bias attractor a plug-in module, i.e., assimilate the training bias during training and be removed in the test stage, and to ease back propagation of the training process.
>
> - - -
> #### **Q2: Running cost analysis.**
> **R2:** Thanks for your constructive comment. We agree that the second-order optimization usually bring extra training computation cost. To verify the efficiency of our L2AC, we quantify the training cost in the table below, where we measure floating point operations per second (FLOPS) using NVIDIA GeForce RTX 3090. Note that L2AC (traditional) denotes the algorithm that unrolls the gradient of the whole classification network to compute the second-order gradient of the bias attractor, just like most gradient-based bi-level optimization algorithm [1,2], while in our work we only unroll the linear classifier to simplify the computation. It can be seen that: (1) our L2AC is much faster than L2AC (traditional); (2) The training cost of our L2AC is comparable to the current SOTA method ABC [4].
>
> **Table R-3:** Training cost analysis on CIFAR-10 and CIFAR-100
> | Method | Params | FLOPS | |
> |  :---- | :----:  | :---:  | :----: |
> | | | CIFAR-10 | CIFAR-100 |
> | FixMatch | 1.47 M | 19.6 iter/sec | 19.6 iter/sec |
> | w/ DARP [3]| 1.47 M | 18.2 iter/sec | 7.5 iter/sec |
> | w/ ABC [4] | 1.47 M | 15.1 iter/sec | 14.9 iter/sec |
> | w/ L2AC (traditional)| 1.48 M | 9.9 iter/sec | 9.7 iter/sec |
> | w/ L2AC (ours)| 1.48 M | 14.2 iter/sec | 13.9 iter/sec |
>
> The results show that the computation cost increment of our proposed L2AC is nearly negligible compared with the baseline models especially considering the significant improvement performance of L2AC. Also, it is worth noting that our proposed L2AC is more efficient than traditional second-order optimization, and we think it is owing to two reasons: (1) the bias attractor only adds a very small number of parameters (about 0.68% of the total number of parameters); (2) To calculate the second-order gradient of these parameters, we only need to unroll the gradient of the linear classifier as shown in Eq. (4).
>
> Note that in the test stage our L2AC requires no extra overhead compared with the baseline model.
> - - -
> #### **Q3: Some notations are ambiguous. In Eq.(2), $\mathbf I$ and $\circ$ seem not to be defined. Also, $|_{\omega^t}^T$ in $G_i$ in Proposition 3.1.**
>
> **R3:** Many thanks for the careful check of details. $\mathbf I$ and $\circ$ denote identity mapping and function composition operator, respectively. $|_{\omega^t}$ represents the gradient operator at $\omega^t$ and superscript $T$ denotes the vector/matrix transpose. We have fixed these issues and marked them in blue in the revised paper.

---

> ### Author Response · Authors · 2022-11-18
> **Response to Reviewer ihj1 (Part 2)**
>
> #### **Q4: t-SNE visualization under other hyper-parameters.**
>
> **R4:** As suggested, we further visualize the t-SNE of training data under other setups, including $\gamma_l=\gamma_u=100$ and $\gamma_l=100,\gamma_u=100$ (reversed) in Fig. 12 and 13 of the revised paper. Compared with FixMatch, our L2AC certainly improves the separability of the tail classes from the head classes. This verifies that the finding in Fig. 5 is not owing to the specific choice of the experimental setups. Please kindly refer to Appendix D.4 for more details.
> - - -
> **Reference**
> >[1] Finn C, et al. Model-agnostic meta-learning for fast adaption of deep networks. In ICML, 2017.
> [2] Ren M, et al. Learning to reweight examples for robust deep learning. In ICML, 2018.
> [3] Kim J et al. Distribution aligning refinery of pseudo-label for imbalanced semi-supervised learning. In NeurIPS, 2020.
> [4] Lee H, et al. ABC: Auxiliary balanced classifier for class imbalanced semi-supervised learning. In NeurIPS, 2021.

---

> ### Author Response · Authors · 2022-12-03
> **Looking forward to your reply and the discussions**
>
> Dear reviewer ihj1,
>
> We sincerely thank you for your efforts in reviewing our paper. Since only a few days remain in the discussion period, please let us know if our responses have well addressed your concerns. If you have any further questions, we are more than happy to address them. Thanks again for your valuable suggestions!
>
> Best regards,
>
> All anonymous authors

---

> > ### Comment · Reviewer_ihj1 · 2022-12-04
> > **Thank you for your response.**
> >
> > Dear authors,
> >
> > Thank you for your response.
> > In terms of the proposed method, if more essential explanations are provided, the value of this paper will increase further.
> > The running cost analysis helps understand the usefulness of the proposed method.
> > Also, the additional t-SNE visualization seems to support the discrimination power of the new method.
> > Overall, I appreciate your reply to my questions.
> >
> > Best regards,
> > Reviewer ihj1

---

> > > ### Author Response · Authors · 2022-12-05
> > > **Thanks for your response and a Bayesian view of our approach**
> > >
> > > Dear reviewer ihj1,
> > >
> > > We sincerely appreciate your constructive suggestions and your time. We are happy to hear that our response addressed most of your concerns.
> > >
> > > Here we further present an intuitive explanation about our L2AC. Traditional pseudo-labeling methods tend to directly fit class-conditional distribution $p(y|x)$ of the biased training data, which makes the linear classifiers easily overfit to training bias. To avoid this, we propose a bias adaptive classifier (the linear classifier + the bias attractor) to fit the biased class-conditional distribution $p(y|x)$, and its two key components play different roles in the training stage: (1) the linear classifier aims to predict the underlying unbiased class-conditional distribution $p(\hat y|x)$; (2) the bias attractor, which can be intuitively regarded as a probability transition model $p(y|\hat y, x)$), targets to assimilates complicated training bias. Towards this goal, the lower-level problem minimizes the empirical risk over the whole biased train data to learn $p(y|x)$ for a better representation learning, and the upper-level loss (imposed on the linear classifier) minimizes the empirical loss over any sampled unbiased (clean and balanced) set for learning $p(\hat y|x)$, making the linear classifier perform well towards each class.
> > >
> > > We hope that these explanations will address your concerns, and we will add these explanations in the final version of our paper. Please kindly let us know if you have any remaining questions. If our response has well addressed your concerns, please could you
> > > consider raising the rating of our work? Thank you very much for helping us further improve our paper!
> > >
> > > Best regards,
> > >
> > > All anonymous authors

---

> > > > ### Comment · Reviewer_ihj1 · 2022-12-08
> > > > **Additional explantions**
> > > >
> > > > Dear authors,
> > > >
> > > > Thank you for providing the new Bayesian view of the proposed approach.
> > > > The new perspective on the proposed method would help readers to understand the paper.
> > > > However, I expected that the explanation in Section 3.2 (in particular, in "The proposed bias adaptive classifier") would be improved. When I first read this paper, I read the figures and explanations several times to understand this paper, but it did not help a lot. To write this review, I found out my personal understanding/interpretation of why this method works. But, I would like to know the easy-to-follow explanations from the authors. The current one is a little bit better than the first one, but I hope that the explanations can be improved.
> > > > For example, there is the line "the bias attractor adopts a nonlinear network which can assimilate complicated training bias in theory due to the universal approximation properties\footnote{In theory, an MLP can approximate almost any continuous function (Hornik et al., 1989).}." This line does not help in understanding the proposed method a lot. Can I remove the bias attractor if I use an MLP for $f^\mathrm{ext}_\theta$?
> > > > Another example is, "we hope the bias attractor could indeed help the linear classifier to learn the unbiased class conditional distribution." This part is difficult to follow. What kind of mechanism helps the linear classifier to learn the unbiased class conditional distribution?
> > > > I picked the two sentences, but it is fine if the overall explanation is sound. That is, it is not necessary to provide long answers to comments to the above two sentences.
> > > >
> > > > Anyway, thank you for the new explanation.
> > > >
> > > > Best regards,
> > > > Reviewer ihj1

---

> > > > > ### Author Response · Authors · 2022-12-08
> > > > > **Thanks for your response and an easy-to-follow explanation**
> > > > >
> > > > > Dear reviewer ihj1,
> > > > >
> > > > > Thank you for your additional time and efforts to carefully read our response. In the following we try to present an easy-to-follow explanation about our approach.
> > > > >
> > > > > Two indispensable factors, namely the proposed **bias adaptive classifier** and **bi-level learning framework**, jointly ensure that our approach can debias for imbalanced SSL tasks.
> > > > > Firstly, since we aim NOT to change the original classification network (the feature extractor $f_{\theta}^{\rm ext}$ + the linear classifier $f_{\omega}^{\rm cls}$) during the test stage, an additional nonlinear network $\Delta f_{\omega}$ (the bias attractor) is thus designed to protect the linear classifier from the training bias in the training stage. Secondly, how to make the bias attractor assimilate the training bias (in other words, how to make the linear classifier exactly fit the unbiased class-conditional distribution) is what the bi-level training does. Specifically, the upper-level loss (imposed on the linear classifier) minimizes the empirical risk over any sampled unbiased (clean and class-imbalanced) set, which makes the linear classifier perform well towards each class, and thus basically fit an unbiased class-conditional distribution.
> > > > >
> > > > > We hope this explanation will address your concerns, and we will revise the final version of our paper accordingly. Please kindly let us know if you have any remaining questions. We really enjoyed discussing with you and sincerely appreciate your constructive suggestions on improving our paper.
> > > > >
> > > > > Best regards,
> > > > >
> > > > > All anonymous authors

---

> > > > > > ### Comment · Reviewer_ihj1 · 2022-12-10
> > > > > > **Thanks for new explanation**
> > > > > >
> > > > > > Dear authors,
> > > > > >
> > > > > > Could you clarify "protect" ("protect the linear classifier from the training bias")? I can understand the feeling and what you mean now, but it would be unclear for readers who read this paper for the first time.
> > > > > > Probably, the assumption of this paper is that the *explicit* imbalanced term (training bias in this paper) due to imbalanced data is decomposed to a balanced term (*unbiased* class-conditional distribution) and the *implicit* imbalanced term. Without the proposed bias adaptive classifier, classifiers, such as vanilla ResNets, suffer from explicit training bias. In other words, classifiers fit both terms.
> > > > > > In contrast, the proposed method lets two classifiers, $f_\omega^\mathrm{cls}$ and $\Delta f_\omega$, fit each term separately to avoid the undesirable effect of the training bias. $f_\omega^\mathrm{cls}$ handles the balanced term, which can be done by minimizing the empirical risk over balanced data. $\Delta f_\omega$ handles the *implicit* imbalanced bias term, which can be done by training $F_{\omega,\phi}$ (=$f^\mathrm{cls}_\omega$ + $\Delta f_\omega \circ f^\mathrm{cls}_\omega$) with the imbalanced training data. In other words, $\Delta f_\omega$ in $\Delta f_\omega \circ f^\mathrm{cls}_\omega$ "absorbs" the *implicit* imbalanced bias term (from another viewpoint, this might be interpreted as that $\Delta f_\omega$ protects $f^\mathrm{cls}_\omega$ from the training bias).
> > > > > > If explanations like the above are given, the mechanism of "protect" might be clarified more. Note that my word choice, such as the imbalanced term, might be inappropriate and strange. Also, my understanding would not be correct.  In this case, please kindly ignore it.
> > > > > >
> > > > > > Best regards,
> > > > > > Reviewer ihj1

---

> > > > > > > ### Author Response · Authors · 2022-12-10
> > > > > > > **Thank you for your very thoughtful review**
> > > > > > >
> > > > > > > Dear reviewer ihj1,
> > > > > > >
> > > > > > > We sincerely appreciate your thoughtful review and your time. You are right that our proposed bias adaptive classifier (the linear classifier equipped with the bias attractor), to some extent, can be thought of as utilizing the linear classifiers and the bias attractors to fit "a balanced term (unbiased class-conditional distribution) and the implicit imbalanced term", respectively. By the way, "the balanced term" may be presented as "the unbiased term", as the training bias includes imbalance bias and pseudo-label bias during training in imbalanced SSL tasks.
> > > > > > >
> > > > > > > Thank you again for this valuable comment which is really helpful to further improve the explanation of our work. We enjoyed discussing with you, and please let us know if you still have any unclear part of our work.
> > > > > > >
> > > > > > > Best regards,
> > > > > > >
> > > > > > > All anonymous authors

---

> > > > > > > > ### Comment · Reviewer_ihj1 · 2022-12-12
> > > > > > > > **Thank you for your response**
> > > > > > > >
> > > > > > > > Dear authors,
> > > > > > > >
> > > > > > > > Thank you for your responses.
> > > > > > > > Although I expected the explanations obtained during the discussion would be presented in the first draft, I admit the authors' effort to improve this paper. I thus increased my score. I hope that the overall explanations will be improved in the final version.
> > > > > > > > One last note that *unbiased* is sometimes confusing because *unbiased* has a special meaning in statistics, e.g., "unbiased estimators."
> > > > > > > >
> > > > > > > > Best regards,
> > > > > > > > Reviewer ihj1

---

> > > > > > > > > ### Author Response · Authors · 2022-12-12
> > > > > > > > > **Thank you for increasing the score**
> > > > > > > > >
> > > > > > > > > Dear reviewer ihj1,
> > > > > > > > >
> > > > > > > > > Thank you for increasing the score and your valuable suggestions which are really conductive to further improve the explanation of our work. We will add the details of the discussion part in the final version！
> > > > > > > > >
> > > > > > > > > Best regards,
> > > > > > > > >
> > > > > > > > > All anonymous authors

---

### Official Review · Reviewer_juK4 · 2022-10-25

**Confidence:** 4
**Correctness:** 3
**Technical Novelty And Significance:** 2
**Empirical Novelty And Significance:** 3
**Recommendation:** 5

**Clarity, Quality, Novelty And Reproducibility:**

Clarity: easy to understand.

Quality: good.

Novelty: fair.

Reproducibility: good.

**Strength And Weaknesses:**

### Pros

1. The paper is easy to follow and well-written.
2. The problem is well-motivated and intuitively makes sense.
3. The reported performance exceeds the baselines significantly.

### Cons

1. The balance loss L_bal is just a subset of the original pseudo-labeling loss L
   (balanced dataset B is a subset of the labeled dataset D_l).
   Thus, if L is minimized, L_bal will also be minimized (if we are overfitting L).
   The effect of introducing a balanced dataset here is unclear.
   If L_bal is a hold-out dataset (instead of "dynamically sampled from the labeled training set"),
   there may be some differences.

2. A much simpler method is to keep the bias-adaptive module fixed when updating other parameters
   and update the bias-adaptive module on the balanced dataset.
   I suggest the authors compare this baseline method and discuss the difference compared with
   the proposed method. The bi-level optimization may be overkill.

3. Theoretical analysis of the convergence of such a bi-level optimizing method is necessary,
   which may help understand the problem in 2.


**Summary Of The Paper:**

This paper proposes a novel imbalanced SSL method with pseudo-labeling.
The authors introduce a bias-adaptive module that can be trained to minimize
classification loss on a balanced dataset with bi-level optimization.

**Summary Of The Review:**

The paper is overall good-looking, but the method does not make much sense to me.

---

> ### Author Response · Authors · 2022-11-18
> **Response to Reviewer juK4 (Part 1)**
>
> Thanks for reviewing our paper and we appreciate your valuable comments. We detail our response below and have revised the paper accordingly. Please kindly let us know whether our response addresses your concerns.
>
> ####  **Q1: Concern on that “the balance loss $\mathcal L^{bal}$ is just a subset of the original pseudo-labeling loss $\mathcal L$” and the two losses will be minimized simultaneously.**
>
> **R1:**  Thanks for your comment. We would like to kindly argue that the balance loss $\mathcal L^{bal}$ is not exactly a subset of the original pseudo-labeling loss $\mathcal L$ for the following reasons.
>
> - The two losses have different inputs even for the same sample. Specifically, $\mathcal L$ is calculated through the whole bias adaptive classifier (the linear classifier + the bias attractor), while $\mathcal L^{bal}$ is computed by the linear classifier of the bias adaptive classifier. Please refer to Fig. 2 of the manuscript for an intuitive visualization.
>
> - The two losses (over $\mathcal B$ vs. $\mathcal D_l$) aim to fit different conditional distributions. This is because $\mathcal B$ and $\mathcal D_l$ follow different class distribution.
>
> Actually, the two losses decrease differently during training. To investigate this, we visualize the training curves ($\mathcal L$ vs. $\mathcal L^{bal}$ in Fig. 10, and $\mathcal L^{bal}$ vs. the training loss only over $\mathcal D_l$ in Fig. 11) in Appendix D.4 of the revised paper. It can be seen that these loss functions converge to values of different magnitudes at different iterations.
> - - -
> #### **Q2: Concern on replacing the sampled balanced set $\mathcal B$ with a hold-out dataset.**
> **R2:**  We agree with the reviewer that replacing $\mathcal B$ with a hold-out set can make the bi-level learning process more robust. However, this will introduce a critical hyper-parameter (how many samples are split from the labeled set) and readily affect the model performance, due to very scare labeled samples given for some classes in imbalanced SSL tasks (e.g., only 15 labeled samples in the most minority class of CIFAR-10/100 under $\gamma_l=100/10$). We investigate this in the table below, and the results show that dynamically sampling is a better choice for our L2AC.
>
> **Table R-1:** Results (bACC / GM) on hold-out balanced set, where \# $n$ denotes $n$ samples per class.
> | Strategies | CIFAR-10 ($\gamma=100$)| CIFAR-100 ($\gamma=10$) |
> |  :----  | :----:  | :---:  |
> | FixMatch | $71.5_{\pm0.71}$ / $66.8_{\pm1.51}$ | $55.1_{\pm0.09}$ / $46.7_{\pm0.53}$ |
> | L2AC (hold-out, \# 2) | $80.6_{\pm0.49}$ / $80.3_{\pm0.53}$ | $56.8_{\pm0.11}$ / $51.9_{\pm0.21}$ |
> | L2AC (hold-out, \# 4) | $81.2_{\pm0.99}$ / $81.0_{\pm0.51}$ | $56.4_{\pm0.19}$ / $51.8_{\pm0.31}$ |
> | L2AC (hold-out, \# 6) | $81.2_{\pm0.17}$ / $80.5_{\pm0.18}$ | $56.3_{\pm0.44}$ / $51.3_{\pm0.62}$|
> | L2AC (hold-out, \# 8) | $81.5_{\pm0.69}$ / $80.7_{\pm0.76}$ |$55.6_{\pm0.10}$ / $49.9_{\pm0.21}$ |
> | L2AC (hold-out, \# 10)| $80.9_{\pm0.72}$ / $80.2_{\pm0.89}$ |$54.8_{\pm0.30}$ / $48.4_{\pm0.51}$ |
> | L2AC (ours, dynamically sampling)| $82.1_{\pm0.57}$ / $81.5_{\pm0.64}$ | $57.8_{\pm0.19}$ / $52.1_{\pm0.31}$
>
> - - -
> #### **Q3: Comparison with the method that alternatively updates bias-adaptive module and other parameters.**
>
> **R3:** As suggested, we have compared the proposed L2AC with the suggested method that alternatively updates the bias attractor and other parameters. For simplicity, we denote this method as AUM.
>
> The experimental setting is as follows. We first tried to keep all the settings the same as our L2AC. However, under this setting, the training of AUM does not even converge, and we found that the training process is sensitive to the learning rate of the bias attractor. After a grid search on this hyper-parameter, we finally set this learning rate as $2e^{-5}$. By contrast, the learning rate of the upper-level loss of our L2AC is $1e^{-4}$. Note that the learning rate of other parameters is kept as 0.002.
>
> In the test stage, we found that AUM cannot decouple the training bias from the linear classifier, since the performance evaluated by the linear classifier is nearly similar to that of random guessing. We thus used the whole bias adaptive classifier (the linear classifier + the bias attractor) for testing. The results are shown in the table below, and our approach surpasses AUM by a large margin.
>
> **Table R-2:** Comparison between AUM and L2AC.
> | Strategies | CIFAR-10| CIFAR-100 |
> |  :----  | :----:  | :---:  |
> | AUM | $78.4_{\pm1.58}$ / $77.5_{\pm1.97}$ | $55.3_{\pm0.73}$ / $50.9_{\pm1.39}$ |
> | L2AC (ours) | $82.1_{\pm0.57}$ / $81.5_{\pm0.64}$ | $57.8_{\pm0.19}$ / $52.1_{\pm0.31}$ |

---

> ### Author Response · Authors · 2022-11-18
> **Response to Reviewer juK4 (Part 2)**
>
> #### **Q4: Theoretical analysis on the convergence.**
>
> **R4:** Many thanks for pointing this out. As suggested, we theoretically analyzed the convergence of our proposed L2AC and we have added a convergence theorem in the Appendix E of the updated submissions. In fact, the training algorithm of our L2AC converges at the rate of $\tilde {\mathcal O}(\frac{1}{\sqrt{T}})$, which meets the convergence results of similar work [1, 2]. Please kindly refer to the revised paper for the detailed proof.
> - - -
> #### **Reference**
> >[1] Ji K, Yang J, Liang Y. Bilevel optimization: Convergence analysis and enhanced design[C]. In ICML, 2021.
> [2] Ren M, Zeng W, Yang B, et al. Learning to reweight examples for robust deep learning[C]. In ICML, 2018.

---

> ### Author Response · Authors · 2022-12-03
> **Looking forward to your reply and the discussions**
>
> Dear reviewer juK4,
>
> We sincerely thank you for your efforts in reviewing our paper. Since only a few days remain in the discussion period, please let us know if our responses have well addressed your concerns. If you have any further questions, we are more than happy to address them. Thanks again for your valuable suggestions!
>
> Best regards,
>
> All anonymous authors

---

> ### Author Response · Authors · 2022-12-07
> **A friendly reminder**
>
> Dear reviewer juK4,
>
> We sincerely thank you for your efforts in reviewing our paper. We have provided corresponding responses and revised  manuscript, which we believe have covered your concerns. As the discussion phase is quickly passing, please let us know if you have any more questions, and we would be happy to address them if needed. Thank you！
>
> Best regards,
>
> All anonymous authors

---

### Author Response · Authors · 2022-11-18
**Summary of Paper Revision**

We sincerely appreciate all the reviewers for their valuable and constructive comments. According to these comments, we have improved the paper and highlighted major revisions in blue. The main changes are summarized as follows:

1. Appending training loss curves in Fig. 10 and 11 to illustrate the convergence of our approach (Appendix D.4)

2. Adding theoretical analysis on the convergence of our training algorithm (Appendix E)

3. Revising Fig. 1 and adding more explanation about training bias in the introduction

4. Adding running cost analysis of our approach (Appendix D.5)

5. Adding feature visualization via t-SNE under different setups in Fig. 12 and 13 (Appendix D.6)

6. Providing descriptions for some missing notations in Section 3

7. Adding a discussion about two missing references in the related work

8. Revising Section 3.2 for more intuitive explanation on each component of the bias adaptive classifier

Since more results and discussions are included in the Appendix, some Appendix indexes have been changed accordingly. We will respond in detail to each reviewer individually. Please feel free to let us know if there is still any confusion.

---

### Decision · Program_Chairs · 2023-01-20

**Decision:**

Accept: poster

**Justification For Why Not Higher Score:**

Incremental and not exciting to other researchers.

**Justification For Why Not Lower Score:**

Thorough work and careful replies to reviewers.

**Metareview: Summary, Strengths And Weaknesses:**

(a) The paper provides a new method to reduce bias towards the majority class in learning from labeled and unlabeled data.

(b) Strengths: The work is thorough and the authors responded to the reviewers carefully.

(c) The work is very incremental. It is a typical ML paper in the sense that it proposes a new algorithm that is a combination of known components, is merely heuristic, and is more complicated than previous methods. The experiments are merely on known datasets and the % gains are small. I don't see a new insight in the paper, or an indication that some task is now feasible that previously was not.

After discussion within the program committee, this paper is accepted for publication at ICLR 2023. Two reviewers are definitely positive. Here are some additional comments that we hope will be useful to the authors and to readers.

- The thoroughness is admirable but ideally would lead to more insight. For example, reviewer juK4 asked for a convergence analysis, which the authors have provided, but this merely says that training converges at the expected rate, which provides no insight about generalization.
- Ideally, the authors would show that some task is feasible with the new method that wasn't before, and they would show success on a novel dataset or application of their own.
- A central topic is class imbalance. For decades, and despite the thouisands of citations of https://dl.acm.org/doi/abs/10.5555/1642194.1642224, many researchers have failed to compare against the simplest approach, namely shifting scores that are obtained using log loss by a constant. The missing comparison here against logit adjustment, "Long-tail learning via logit adjustment" https://arxiv.org/abs/2007.07314 is in this vein (none of the reviewers are connected to this paper).

Wrt insights, here are comments extracted from the original reviews and then from replies by reviewers. Score 5: "The paper is overall good-looking, but the method does not make much sense to me." Score 7 raised to 8: "The current explanation of the proposed method is not really clear." Later: " When I first read this paper, I read the figures and explanations several times to understand this paper, but it did not help a lot. To write this review, I found out my personal understanding/interpretation of why this method works. But, I would like to know the easy-to-follow explanations from the authors. The current one is a little bit better than the first one, but I hope that the explanations can be improved." Score 8: "The motivation of the proposed bias adaptive classifier is still a bit vague to me" Hopefully the final version of this paper will address this feedback successfully.


**Note From Pc:**

if the above contains the word "oral" or "spotlight" please see: "oral" presentation means -> notable-top-5% and "spotlight" means -> notable-top-25%. As stated in our emails, we are disassociating presentation type from AC recommendations

**Summary Of Ac-Reviewer Meeting:**

No meeting.